# Optimizing spatial equity of urban park cooling services: Integrating landscape metrics with K-means and PSO algorithms in Nanchang, China

Youqiang Zhao[1], Liu Pinyi[2], Gong Peng[3]*, Zhang Jian Ping[4]*

1 Department of Gardening and Art, Jiangxi Agricultural University, Nanchang, Jiangxi Province, China,
2 School of Resources & Environment, Nanchang University, Nanchang, Jiangxi Province, China,
3 Correspondence: Jiangxi Agricultural University, Jiangxi Society of Landscape Architecture, Nanchang, Jiangxi Province, China, 4 Department of Gardening and Art, Jiangxi Agricultural University, Nanchang, Jiangxi Province, China

* 28505587@jxau.edu.cn (GP); zjp112@jxau.edu.cn (ZJP)

## Abstract

Urban parks and green spaces (UPGS) provide critical cooling services to mitigate urban heat islands, yet their equitable distribution remains poorly addressed. This study integrated landscape metrics with spatial optimization algorithms to quantify and enhance the cooling equity of UPGS in Nanchang, China—a city experiencing severe heat stress. Using Landsat 8TIRS data (2021), we analyzed 85 UPGS to extract cooling indicators (LST, PCI, PCA, PCG) and correlated them with landscape composition (area, perimeter, impervious/green/water coverage) and pattern indices (PD, LPI, etc.). Network analysis based on road networks and 3,024 settlements evaluated accessibility to cooling ranges. Results showed 71 UPGS exhibited significant cooling effects ($P < 0.05$), with optimal thresholds at 60 hm$^2$ area and 3 km perimeter. Water coverage was most strongly associated with lower LST ($R^2 = 0.4284$), while complex green patch morphology extended cooling distance. Crucially, only 71.2% of residents could access cooling services within a 15-min walk, revealing severe suburban disparities (e.g., 59.1% coverage outside Second Ring Road vs. > 73% intra-city). To address gaps, we combined K-means clustering (identifying 18 optimal UPGS additions) and Particle Swarm Optimization (locating placements prioritizing suburban demand). This framework bridges micro-scale UPGS design (e.g., maximizing water bodies) and macro-scale algorithmic spatial planning, offering actionable strategies for thermally equitable cities.

## 1. Introduction

The clustering of the urban population has become a widespread characteristic of global development [1]. As of 2020, more than 56% of the global population resides

**Data availability statement:** All relevant data are within the paper and its Supporting Information files.

**Funding:** The author(s) received no specific funding for this work.

**Competing interests:** The authors have declared that no competing interests exist.

in urban areas, occupying merely 0.7% of the Earth's land area [2,3]. The significant imbalance between population size and permanent land utilization has exacerbated human-environment conflicts. This imbalance manifests acutely in challenges related to equitable access to public resources and the exacerbation of urban climate issues, such as the urban heat island (UHI) effect [4]. The UHI effect, a typical extreme urban climate phenomenon, significantly endangers the urban atmospheric and water environments, as well as residents' health, by modifying heat transfer between the atmosphere and the surface [5–9]. Extensive research has demonstrated the valuable ecological benefits of urban parks and green spaces (UPGS) as an important part of the city's green infrastructure while supporting residents' outdoor recreation [10–13]. Within UPGS, vegetation lowers local surface temperature through shading and evapotranspiration (plant transpiration), while water bodies contribute primarily through evaporation. "This localized cooling phenomenon is commonly termed the "cooling island effect" (CIE) [14–16].

Research indicates key landscape composition metrics influencing UPGS CIE include area, perimeter, integration index, impervious and green surface coverage, and water body coverage [17–19]. For instance, the average surface temperature within UPGS generally exhibits a positive correlation with water bodies and green coverage, while demonstrating a negative correlation with impervious surfaces [20–23]. Additionally, the intricacy of green space and water body morphology can amplify the CIE of UPGS [21]. The relationship between UPGS CIE and both area and perimeter is nonlinear, characterized by thresholds beyond which cooling benefits diminish [23,24]. In addition, the landscape pattern index in landscape ecology possesses a further description of the urban cold island pattern [25–28]. For example, the degree of aggregation and complexity of water body patches enhances the cooling benefits, and the degree of separation between impervious surface patches and green space patches is also positively correlated with the cooling benefits under certain thresholds [29–32].However, despite substantial research on the relationship between UPGS landscape composition and cooling indicators [33–35], two critical gaps persist. First, the issue of equitable spatial distribution of UPGS cooling services has received comparatively less attention [36]. Second, there is a lack of integrated frameworks that translate empirical findings on cooling mechanisms into actionable spatial planning solutions.

The CIE provided by UPGS constitutes a valuable public ecosystem service, yet its distribution is often spatially uneven across urban communities [37]. This unequal distribution pattern often correlates with decisions made by local governments and prevailing economic conditions [38,39]. A significant portion of urban residents, particularly those exposed to extreme heat events, being immediate beneficiaries of this resource, should garner greater attention towards its equitable distribution [40]. Accessibility, as a key reference for environmental justice issues, has been more maturely constructed as a correlation between the supply of UPGS resource services and the demand of the population [40]. Directing attention to urban residents' access to the cooling range of UPGS can better guide the provision of cooling services and foster environmental justice [41]. Established methods for evaluating UPGS

accessibility encompass the minimum distance method, buffer analysis, gravity model, network analysis, and two-step floating catchment area. The minimum distance method and buffer analysis overlook the actual road network [42], while the gravity model and two-step floating catchment area consider various factors including attraction, supply, demand, and spatial friction. However, they involve complex data requirements and determining the resistance coefficient is challenging [43,44]. Conversely, the network analysis method revolves around supply points and calculates service areas based on the actual road network within specific time or distance parameters. It prioritizes residents' time and cost constraints, aligning more consistently with an assessment of the fairness of the CIE [40,45]. Simultaneously, analyzing the spatial distribution pattern of UPGS cooling services aids in optimizing the subsequent spatial layout of UPGS cooling services. Previous research on optimizing UPGS resource fairness employed genetic algorithms to attain the most equitable UPGS layout [46], and the siting of ecological land through the combination of multi-intelligentsia and ant colony algorithms [47]. However, these two methods entail complex parameter settings and are more prone to local optimal solution issues [48]. To address these limitations, this study employs a combined approach utilizing the K-means clustering algorithm (KMS) and the Particle Swarm Optimization algorithm (PSO). This framework aims to identify cooling service equity by determining the optimal planning strategy for new UPGS. Specifically, the former (KMS) calculates the optimal number of new UPGS required within the identified blind zones, while the latter (PSO) identifies their precise spatial locations.

Both quantifying and optimizing the UPGS CIE and its equitable distribution are crucial given the intensifying urban thermal environment [49]. This study focuses on Nanchang City, the capital of Jiangxi Province and a major economic hub in East China, experiencing population growth and urban expansion in recent decades. As per the seventh national census, Nanchang's resident population has reached 6,255,000, showing an increase of 1,123,000 from the previous census, with an urbanization rate nearing 80% [50]. Rapid urbanization and economic development in Nanchang, often involving the conversion of agricultural and forest land, have not only yielded economic growth but also intensified urban climate extremes [51]. Statistics indicate that between June and August 2022, Nanchang City experienced 55 days with temperatures exceeding 35°C including 12 days with extreme temperatures above 39°C (https://www.tianqi24.com/), earning Nanchang the title of "China's four major hot stove cities." Therefore, investigating the spatial patterns and drivers of UPGS CIE in Nanchang is crucial for informing strategies to enhance urban thermal comfort and livability.

In summary, this study develops and applies a novel, closed-loop analytical framework for Nanchang's UPGS that integrates microscopic analysis with macroscopic spatial planning to bridge the identified research gaps.This is achieved through four logically consecutive steps:1) Quantify the CIE of UPGS and its correlation with landscape metrics to identify key cooling drivers (LST, PCI, PCA, PCG); 2) Assess the spatial equity of residents' access to cooling services to diagnose priority areas in need; 3) Proposing an optimized spatial plan for new UPGS using KMS and PSO algorithms, thereby synthesizing the insights from cooling drivers and equity gaps to maximize coverage; and 4) Discussing how the micro-level design implications and macro-level planning strategies, derived from the previous steps, can be combined to guide thermally equitable urban development (Fig 1). The principal contribution of this work lies in this translational framework that explicitly links empirical understanding of cooling mechanisms to algorithmic optimization for equitable spatial planning, offering a replicable model for cities facing similar heat equity challenges.

## 2. Materials and methods

### Study area

Situated in East China, Nanchang City serves as the capital of Jiangxi Province, spanning longitude 115°27'-116°35'E and latitude 28°10'-29°11'N (Fig 1). Covering an area of 1005 km², Nanchang comprises seven districts: Donghu, Xihu, Honggutan, Qingyunpu, Qingshan Lake, Xinjian, and Nanchang County. The city is delineated by the First Ring Road and Second Ring Road into core urban, inner suburban, and outer suburban zones (Fig 2). Nanchang experiences a typical

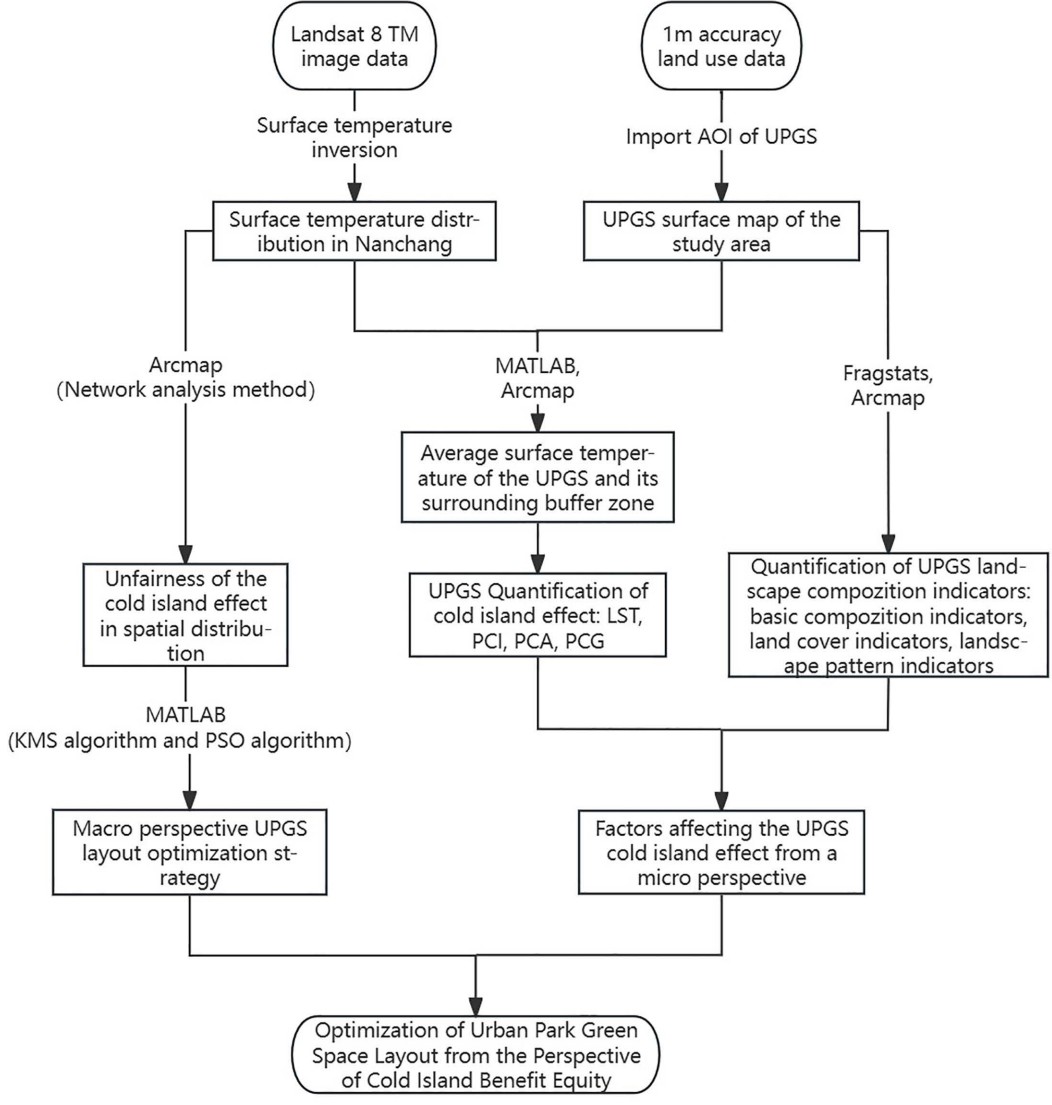

**Fig 1. Conceptual framework for assessing and planning the CIE in urban parks and green spaces.**

subtropical monsoon climate characterized by abundant heat and rainfall. Nonetheless, substantial annual variations in monsoon strength result in significant temperature fluctuations, leading to frequent meteorological disasters such as high temperatures, droughts, cold spells, and related damages. As per the data from the Nanchang Temperature Network, the average daily maximum temperature over the last three years has been 24°C with an average annual rainfall of 956.7 mm. Moreover, there is a notable trend of increasing occurrences of hot weather (https://www.tianqi24.com/).

## Data sources and processing

**2.1.1. Remote sensing image data.** Surface temperature data for Nanchang City was obtained from Landsat 8TIRS remote sensing data captured on September 25, 2021, along path 121, row 40, with near-zero cloud cover (<1%). On the day of satellite transit, Nanchang City experienced temperatures ranging from 26°C to 36°C, as reported by the China Meteorological Data Network (http://data.cma.cn/).

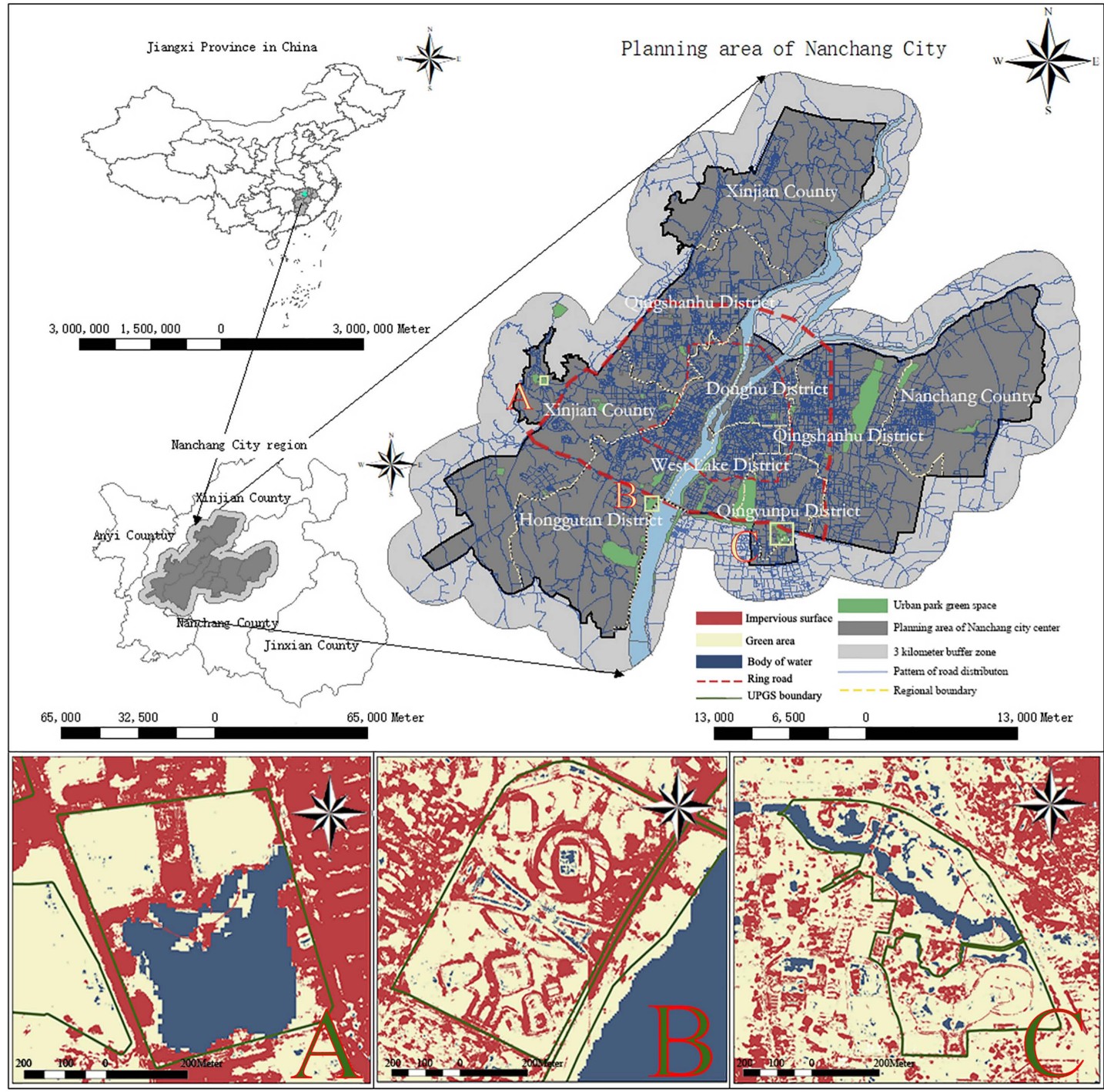

**Fig 2. Geographic location of the study area and spatial distribution and land use of urban park green spaces (The base map for this figure contains data from OpenStreetMap (available under the Open Database License) and Landsat 8TIRS).**

Land data were obtained from the initial national land cover map (SinoLC-1) with a 1m resolution in China [52], and the original land use data were partially corrected by manually visually interpreting 0.6m-resolution high-definition remote sensing imagery, focusing on UPGS boundaries and land cover types within UPGS, to maximize the restoration of the most recent original site appearance.

**2.1.2. UPGS indicator extraction.** Initially, the Point of Interest (POI) directory of UPGS in Nanchang City was accessed using Python, followed by utilizing the Baidu map application programming interface to extract the Area of Interest (AOI) UPGS data, which was then imported into the map. Considering that the use of UPGS by residents near the boundary of the study area is not limited to the internal area, the UPGS in the 3-km buffer zone on the periphery of the study area was also set as the study object, and a total of 85 UPGS were counted and crawled. With reference to the "Nanchang City Green Space System Planning (Revision) (2015-2020)", we classified the 85 UPGSs in the study area into four categories by the area parameter of the UPGSs: comprehensive (≥25 hm$^2$, the number of 20), city-wide ([5–25] hm$^2$, the number of 41), regional ([2 –5] hm$^2$, the number of 13), and community (<2 hm$^2$, the number of 11). Since some of the UPGS scales are small, the UPGS land use types are classified into three categories: impervious surfaces, greenery, and water bodies, which were reclassified into three categories: impervious surfaces (including roads and buildings), greenery (including forests, grasslands, croplands, and unused land), and water bodies [25]. The land use information after the reclassification of the three UPGSs is shown in Fig 2.

To quantify the landscape composition indexes of UPGS, ArcMap 10.6 software (ArcGIS Desktop 10.6, https://www.esri.com/) was utilized to extract the area, perimeter, and shape index of UPGS as fundamental composition indexes. Furthermore, the reclassified impervious cover, green cover, and water body cover were extracted to serve as park surface cover indexes. Based on prior study, six landscape pattern indices were considered: patch density (PD), landscape patch index (LPI), landscape shape index (LSI), division index (DIVISION), split index (SPLIT), and aggregation index (AI) [53] (Table 1).

**2.1.3. Calculation of UPGS CIE Indicators.** Previous studies have compared three methods for land surface temperature inversion, namely, the radiative transfer equation method, the single-channel algorithm, and the split-window algorithm, have been compared, and it is found that the surface temperature data inverted by the radiative transfer equation method are the closest to the measured values [22,54]. Therefore, this study also adopts the radiation equation transmission method to invert the surface temperature with Landsat 8TIRS remote sensing data from Nanchang City in the software ENVI 5.6 (The Environment for Visualizing Images, https://envi.geoscene.cn/appstore/), and the results are shown in Fig 3. To quantify the cooling effect of UPGS, a 3000m buffer surrounding 85 UPGS was generated in ArcMap 10.6.This buffer distance was selected as a conservative threshold to fully encompass the potential cooling range of all parks, thereby ensuring the complete capture of the cooling gradient for each UPGS, including those with the largest observed cooling distances [55]. The buffer was then overlaid with the surface inversion result layer to derive the surface

Table 1. Meaning of landscape pattern index [53].

| Landscape pattern index | | Meaning | Landscape pattern index | | Meaning |
|---|---|---|---|---|---|
| The patch density | PD | A measure of the population density of individual patches within an area, used to assess the complexity and diversity of ecosystems. | Division index | DIVISION | Refers to the degree to which the landscape is fragmented, reflecting the complexity of the spatial structure of the landscape. |
| Landscape patch index | LPI | The proportion of total landscape area occupied by the largest patch in the patch type. | Split index | SPLIT | Refers to the separation of the distribution of different patch numbers of individuals in a given landscape type. |
| Landscape shape index | LSI | Means the total edge length of the relevant patch type divided by the minimum possible value of the total edge length. | Aggregation index | AI | Measures the degree of aggregation of patches in the landscape, i.e., connectivity and distance between neighboring patches. |

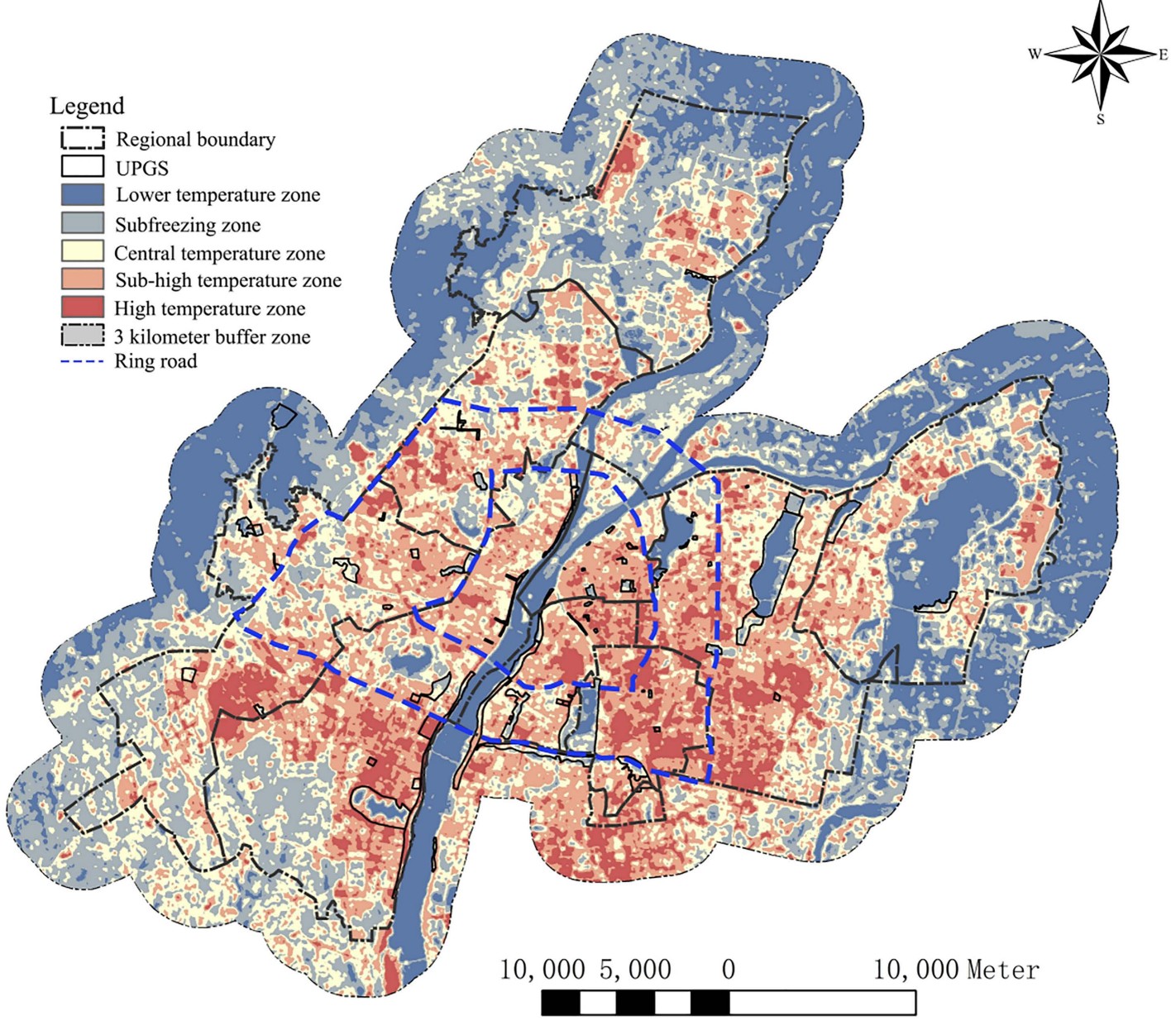

**Fig 3. Surface temperature of the study area (The base map for this figure contains data from OpenStreetMap (available under the Open Database License) and Landsat 8TIRS).**

temperature within the buffer zone. Within MATLAB 2023a, concentric 30m-wide buffer rings were generated outward from each UPGS boundary up to 3000m. The mean LST within each ring was calculated, generating an LST profile (cooling curve) for each UPGS [56] (Fig 4). Following the principle of decreasing boundaries, the surface temperature within the buffer increases progressively with distance from the UPGS, indicating a decrease in cooling efficiency [56]. We evaluated the cooling capacity of the whole UPGS in terms of the first turning point of the temperature profile based on previous study, which was objectively identified as the breakpoint where the cooling gradient changed most

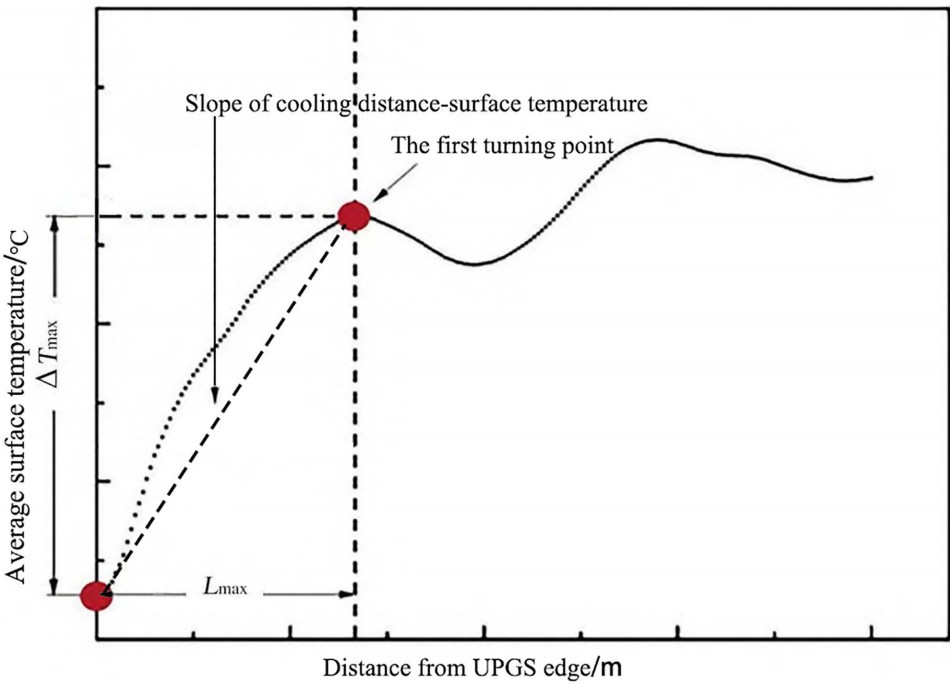

**Fig 4. Schematic diagram of UPGS cooling curve.**

significantly, using a piecewise linear regression algorithm.. The distance from the boundary of the UPGS to the turning point was defined as the maximum cooling distance ($L_{max}$) for the cold-island effect, and quantified the cold-island effect by the cooling amplitude (PCI), the cooling distance (PCA), and the cooling efficiency (PCG) [57]. PCI is defined as the difference between the surface temperature at the first inflection point and the mean surface temperature of the UPGS; PCA is defined as the maximum buffer affected by the CIE of the UPGS, i.e.,the linear distance from the UPGS boundary to the inflection point; and PCG stands for the slope of the curve, which is defined as the mean cooling intensity per unit of distance, i.e., PCI/$L_{max}$ [58].

 **2.1.4. Statistical analysis of CIE data.** To investigate the impact of different UPGS landscape composition indicators on the CIE, Pearson or Spearman correlation analysis was performed in SPSS 23.0, as appropriate based on the Kolmogorov-Smirnov normality test results, ensuring the robustness of correlation inferences. Variables showing significant correlations (e.g., $P < 0.05$ or $P < 0.01$) were further analyzed using regression modeling in Origin 2022 to explore potential predictive relationships. Furthermore, to ensure the robustness of the statistical findings, the variance inflation factor (VIF) was calculated for all independent variables, with values below 5 indicating no severe multicollinearity. A Bonferroni correction was applied to account for multiple comparisons. As most results were nominally significant ($P < 0.05$) but did not meet the stringent corrected threshold ($P < 0.005$), they are cautiously interpreted as indicating potential trends in the discussion of our findings.

## Analysis of CIE intensity direction

In addition to the aforementioned frequent academic references to the influence of the three major constituent indicators of UPGS on the CIE, determining the direction of the intensity of the CIE is similarly beneficial in guiding the construction of a more cooling-efficient UPGS and better serving urban residents [59,60]. The direction of the strong effect of the cold island was determined by using the Hot and Cold Spot Analysis tool and the Standard Ellipse Difference tool in ArcMap.

Several representative UPGSs are selected, and the hot and cold spots inside the UPGSs and buffer zones are calculated and the standard ellipse is generated, and then the factors affecting the direction of the CIE of the UPGSs are determined by comparing the ellipse pointing direction and the flatness of the two, which can provide suggestions for the subsequent layout of the cold spots in the UPGSs.

## Accessibility analysis of the CIE and its layout optimization

Accessibility refers to residents' ability to spatially reach the cooling range of Urban Parks and Green Spaces (UPGS), directly quantifying the cooling benefits provided. However, as the cooling range is typically spatially extensive and morphologically complex, and its service (spatial accessibility) does not involve supply and demand in terms of quantifiable resource volumes, the assessment primarily focuses on residents' time and cost constraints during travel. Therefore, network analysis serves as the core method for evaluating accessibility to the UPGS cooling effect [56], as it realistically simulates travel along the road network to reach this cooling range while incorporating time and distance constraints.

This study's network analysis method is rooted in Nanchang City's actual road network. It simulates the traffic resistance encountered by residents walking on foot, reconstructing the time residents spend accessing the UPGS's cooling range. Road data for Nanchang City was acquired by accessing the OpenStreetMap website (https://www.openstreetmap.org/copyright), and then imported into ArcMap for subsequent analyses after topology checking, road network matching, data correction, and filtering. Settlement data were gathered from Anjuke's website (https://wuhan.anjuke.com/),  one of China's largest second-hand housing trading platforms. After filtering and correcting, 3024 settlements were obtained, and the community population count was derived by multiplying the number of households in the community by the administrative unit's population (Nanchang City Bureau of Statistics). Additionally, considering traffic light conditions and variations in walking ease across road types, residents' walking speed was set at 1 m/s on main roads and 1.2 m/s on internal roads, branch roads, and secondary roads [61]. The intersection of the PCA service boundary and each UPGS road was designated as the cooling service's supply point, establishing service zones accessible within 5, 10, and 15-minute walking times. The equity level of cooling services in Nanchang was evaluated by counting the number of residents in each service zone.

Cooling service supply-blind zones are addressed by integrating the KMS and PSO. KMS clusters settlements lacking cooling coverage, with its interpretability, simplicity, and scalability for large datasets [62] enabling determination of the optimal cluster count (*k*)—representing needed UPGS locations—using the elbow method based on Euclidean distance metrics. Seamlessly integrated, PSO then spatially optimizes siting by initializing particles at KMS cluster centroids, where each particle encodes a candidate UPGS location [63]. The objective function minimizes total Euclidean distance from uncovered settlements to their nearest new UPGS, maximizing coverage efficiency. Particles iteratively update positions using personal best (pbest) and global best (gbest) values until convergence or iteration limits (e.g., 2,000 cycles) are met, ultimately outputting the gbest coordinates as optimal pre-siting locations for *k* new UPGS.

## 3.  Analysis of results

### Temperature characterization of the city as a whole and UPGS

The surface inversion results in Nanchang exhibit a minimum temperature of 30.2°C and a maximum temperature of 57.5°C. The mean-standard deviation method is employed to categorize the inversion results into five classes (Fig 3). Among these classes, the high-temperature zone is predominantly concentrated in the lower-right corner of the first and second ring roads and the lower-left corner of the second ring road. This area coincides with the dense distribution of industrial zones in Nanchang. The sub-high-temperature zone covers the region within the first and second ring roads and the edge of the second ring road, closely overlapping with the distribution of settlements in Nanchang. The low-temperature zone and the sub-low-temperature zone are primarily situated in the urban fringe areas, large water

bodies, and dispersed UPGS. The high-temperature areas where residents congregate are geographically distant from the primary low-temperature areas of the city. The dispersed UPGS in Nanchang serves as the primary source of cooling services that most residents can enjoy at a low cost.

Statistically, among the 85 UPGS in Nanchang City, 71 exhibit significant CIEs (P<0.05) while most of the 14 UPGS without CIEs are dominated by large areas of impervious surfaces or grasslands, lacking significant plant shading and water bodies, which is also consistent with previous studies [64,65]. Some of these 71 UPGS are influenced by surrounding environmental cooling services. Their exclusion minimizes errors in correlational studies of UPGS CIEs, not hindering subsequent explorations. Post-screening, the 56 retained UPGS displayed an LST of 40.281°C, which is much lower than the LST of 41.253°C for the whole of Nanchang city. Among them, the LST of the highest UPGS was 45.302°C, and the lowest LST was 34.392°C, with a difference of more than 10°C. This also indirectly demonstrates that the differences in the landscape composition indexes of the UPGSs can significantly affect the CIE. Further analyses revealed that UPGS' PCI ranged from 0.02°C to 3.096°C (average 1.054°C), PCA spanned from 30m to 1380m (average 396m), and PCG ranged from 0.0012°C/m to 0.006°C/m (average 0.0034°C/m). Subsequently, the data were statistically analyzed using the park type of the UPGS as the X-axis and the cooling distance (PCI) and cooling amplitude (PCA) of the UPGS as the Y-axis to observe the dispersion of the data (Fig 5). It was found that larger comprehensive UPGS showcased the highest PCI and PCA, while smaller community-type UPGS demonstrated a more consistent CIE (Fig 5). In addition, the PCA gap between regional, citywide, and composite UPGSs was not significant, suggesting that exploring landscape composition metrics other than area is necessary for the enhancement of CIE in UPGS.

## Correlation between UPGS landscape composition indicators and CIE

Before conducting the correlation analysis of the CIE, the distribution of UPGS landscape composition indicators and quantitative CIE indicators were assessed. The Kolmogorov-Smirnov test revealed that LST, PCA, PCI, impervious surface

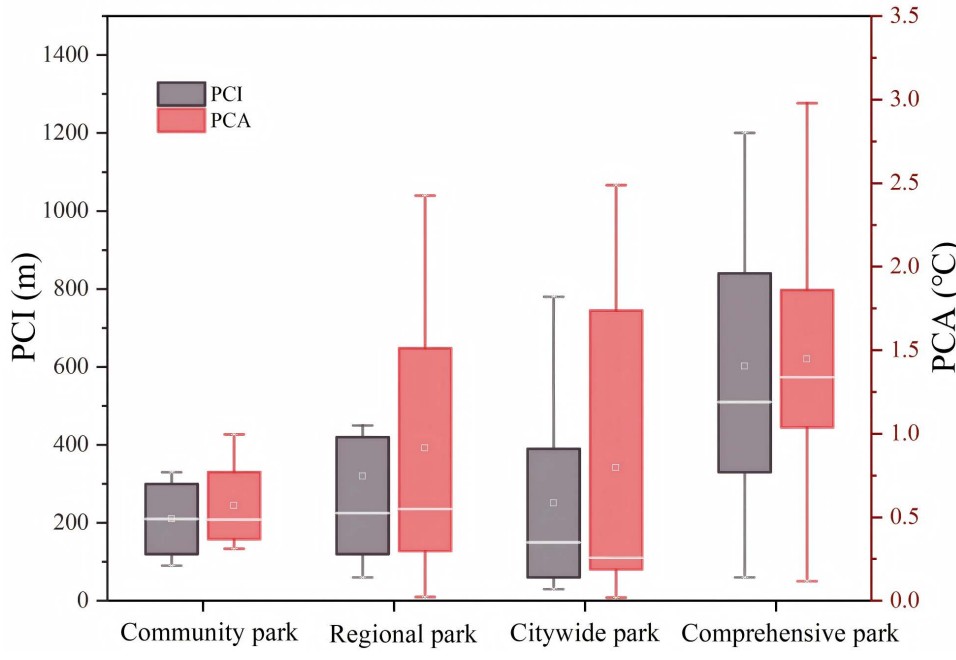

**Fig 5. Box line plot of UPGS CIE.**

PD, green space PD, impervious AI, and green space AI displayed normal distributions and correlation analyses were conducted using Pearson. Conversely, the remaining landscape composition indicators and quantitative CIE indicators exhibited non-normal distributions. Spearman correlation analysis was employed for the non-normally distributed data [66].

**3.1.1. Analysis of UPGS basic composition indicators and surface coverage indicators.** The correlation analysis between UPGS basic constituent indexes, surface cover indexes, and the quantitative CIE indexes is depicted in Fig 6. The analysis revealed significantly negative relationships (P < 0.01) between LST and UPGS area, perimeter, shape index, and water body coverage. However, no significant correlation was observed between LST and impervious surface coverage and green coverage. PCI showed a significant (P < 0.05) negative correlation with UPGS perimeter, impervious surface coverage, and water body coverage (P < 0.01), while its correlation with impervious surface coverage and green coverage was non-significant. PCA displayed a substantial positive correlation (P<0.01) with UPGS area, perimeter, shape index, and water body coverage, alongside a significant negative correlation (P<0.05) with impervious surface coverage. Conversely, PCG exhibited no significant correlations with the basic composition and surface cover indicators. Crucially, it was found that green coverage demonstrated no statistically significant correlation with any of the three quantitative CIE indicators (PCI, PCA, and PCG). This discrepancy may be attributed to the composition of vegetation types within Nanchang's UPGS. It is plausible that a considerable portion of the 'green coverage' is comprised of lawns or shrubs, which have a markedly lower cooling effect through evapotranspiration compared to dense canopy trees, as evidenced by studies in similar climatic contexts [31,67], suggests that within the specific context of Nanchang's UPGS, the mere presence of green area is not a reliable predictor of cooling efficiency, and underscores that perimeter has a more determinant role than green coverage among the basic landscape metrics [68].

To elucidate the quantitative relationship between UPGS basic constituent indicators, surface cover indicators, and the CIE, fitting models are constructed based on correlation analysis. Observing a-d in Fig 7, the UPGS area showed the strongest linear relationship with LST among the basic composition indicators, represented by a fitting coefficient $R^2$ of 0.4284, followed by UPGS perimeter, shape index, and water body coverage. LST diminishes as the UPGS area gradually increases, and the slope levels off beyond a specific threshold value. In conjunction with Fig 7h, which relates to another area metric, it can be seen that the impact threshold for the area of the UPGS is 60 hm², beyond which both the LST decrease and the PCA increase taper off to zero. Evident from Fig 7's e-g, PCI has the highest degree of correlation with UPGS perimeter, followed by impervious surface coverage and water body coverage. Considering b, e, and i in Fig 7, the

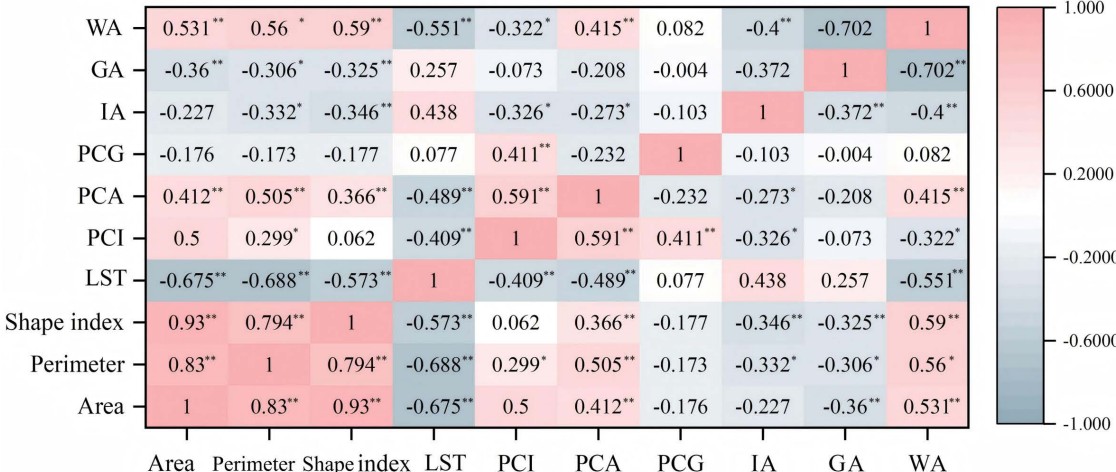

**Fig 6. Heat map of significance analysis of UPGS CIE with basic constituent indicators and surface cover indicators.**

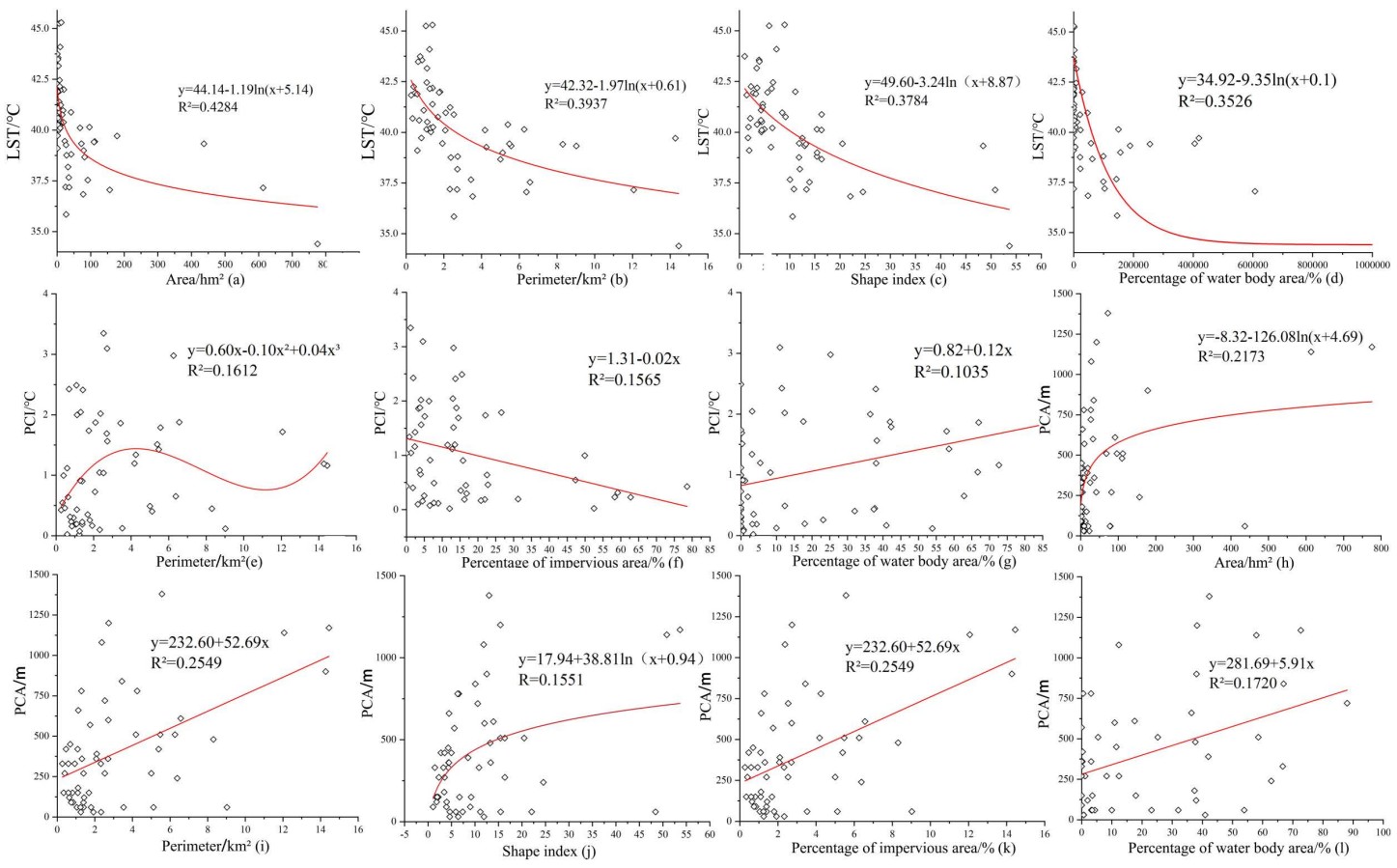

**Fig 7. Correlation factor analysis of the UPGS CIE with basic constituent indicators and surface cover indicators (a-l).**

maximum cold island benefit is observed at a UPGS perimeter of 3 km. Lastly, from the h-l observations in Fig 7, UPGS area, perimeter, and impervious surface coverage affect PCA similarly, while the UPGS shape index has a lesser impact in relation to water body coverage. Combining the other fitted models for impervious surface coverage in Fig 7, it is evident that lower impervious coverage significantly benefits the CIE of the UPGS.

From the above analyses, it can be concluded that the CIE of UPGS isn't singularly determined but rather influenced by various interrelated factors, each with different impacts, and this conclusion has been confirmed by the total number of scholars [21,53,69]. By comparing these factors, the landscape composition indicators with the greatest influence on the CIE are identified, facilitating the optimization of the UPGS layout. For instance, a UPGS with an area of 60 hm² and a perimeter of 3 km falls within the threshold range for generating a stronger CIE. Moreover, reduced impervious surface area and increased water body area correlate with lower LST and larger PCA.

**3.1.2. Landscape pattern index analysis.** Besides the analysis of UPGS basic composition and surface cover indexes, exploring the relationship between UPGS landscape pattern indexes and ambient temperatures is valuable [56,70]. In this study, six landscape pattern indexes, namely PD, LPI, LSI, DIVISION, SPLIT, and AI, were introduced. Retained indices were extracted into Origin to build a correlation model (Fig 8), excluding landscape pattern indices with non-significant correlations in SPSS. The results revealed positive correlations between impervious surface PD, green PD, water body PD, and LST in Fig 8a. Impervious surface PD had the steepest slope, followed by green PD,

   

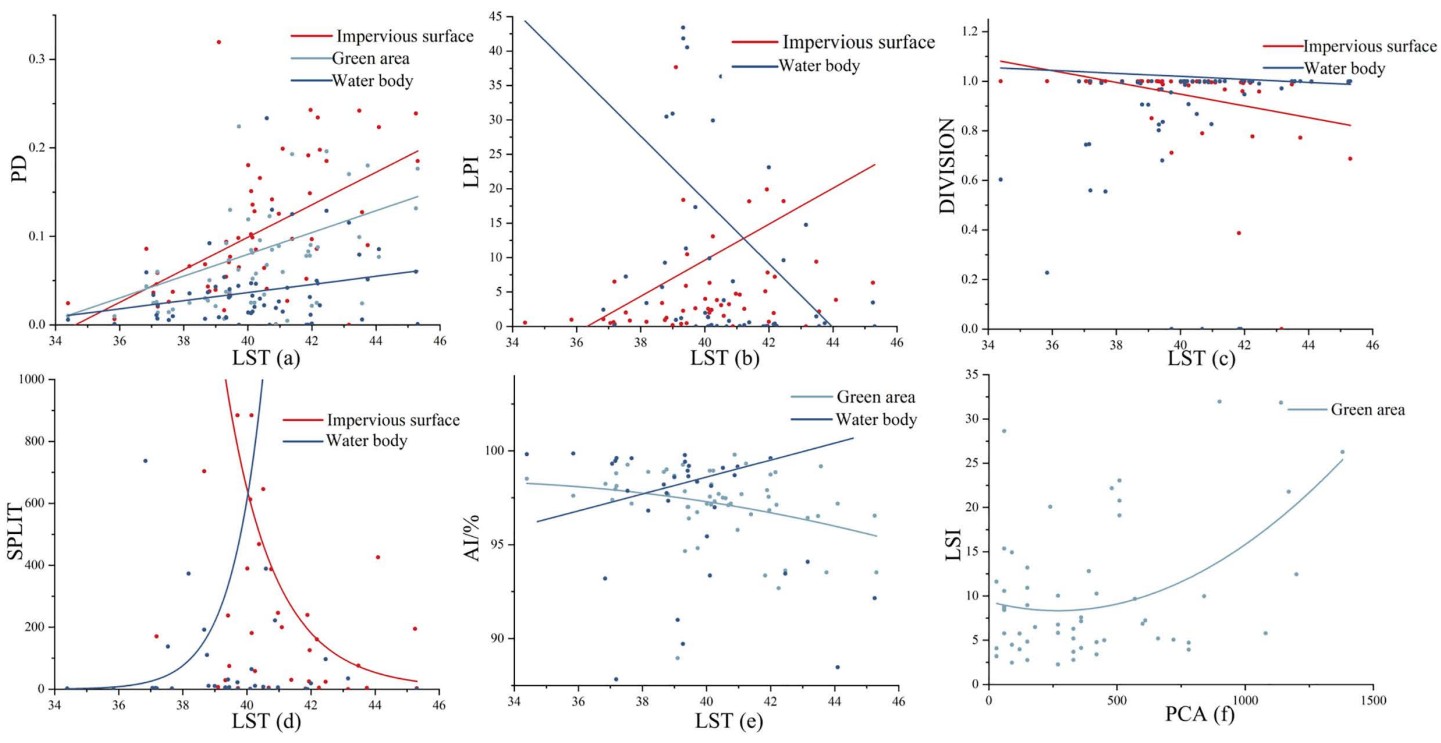

**Fig 8. Factor analysis of the correlation between UPGS CIE and landscape pattern indicators (a-f).**

and water body PD. This suggests that higher patch densities of impervious surfaces and greenery bodies contribute to higher internal UPGS temperatures, with water bodies having the smallest impact. Fig 8b displays a positive correlation between impervious surface LPI and LST, while water body LPI had a negative correlation with LST. This indicates that lower percentages of impervious surfaces and higher percentages of water bodies within UPGS result in lower internal temperatures. In Fig 8c, both impervious surface DIVISION and water body DIVISION negatively correlated with LST, but the correlation for water body DIVISION was weaker. This suggests that higher fragmentation and more divisions within impervious surface patches enhance the CIE, while the degree of division among water body patches makes a limited contribution. In Fig 8d, impervious surface SPLIT showed a negative correlation with LST, while water body SPLIT exhibited an exponential positive correlation with LST. This implies that finer-grained water body patches and greater spatial separation result in higher internal UPGS temperatures, but this negative impact gradually diminishes beyond a certain threshold. On the other hand, higher spatial aggregation of impervious surface patches favors the UPGS's CIE, also exhibiting a threshold effect. In Fig 8e, green AI negatively correlated with LST, while water body AI showed a positive correlation. This indicates that higher dispersion of water body patches or greater aggregation of greenery patches and stronger connectivity promotes the CIE of UPGS, confirming the reliability of the water body SPLIT conclusion. In Fig 8f, green LSI and PCA primarily exhibited a positive exponential correlation. Within a specific threshold, the complexity of greenery patch morphology had no significant correlation with UPGS PCA. And breaking through a certain threshold, higher complexity in greenery patch morphology is more conducive to expanding the CIE's scope in UPGS.

## Analysis of the direction of UPGS CIE intensity

Determining the direction of UPGS CIE intensity is crucial for optimizing UPGS layout and resident distribution [59]. We selected three representative UPGSs from the 56 available: Aixi Lake Park, a large UPGS with a water source; Meihu

Park, a medium-sized UPGS with a water source; and Yangming Park, a small UPGS without a water source. We employed the standard deviation ellipse tool for directional analysis of cold island intensity [60,71]. In Fig 9a, the distribution of cold spots within Aixi Lake Park is mainly contributed by water bodies, and their distribution in the buffer zone mirrors that in the green space. The flattening rate of ellipse 1 in the park is much smaller than that of ellipse 2 in the buffer zone, indicating a more distinct CIE direction within the park. Ellipse 1 and 2 share a similar orientation, aligned with the long axis of the park's water body, highlighting the larger water body's pivotal role in determining the cold island intensity direction within the park and its buffer zone. Fig 9b reveals that Meihu Park's smaller water body area, ellipse 1, aligns with the long axis of the water body, while ellipse 2's orientation leans more towards the water body between the green space and water body cold spot effects. Ellipse 3 in the park is considerably less oblate than ellipse 4 in the buffer zone. Once again, the predominance of water bodies compared to green spaces in determining the direction of the UPGS cold island strength is confirmed. Fig 9c shows that cold spots in waterless Yangming Park are mainly influenced by green space inside the park, while those in the buffer zone are more affected by building shadows than greenery. Ellipses 5 and 6 exhibit similar oblateness. In the park, the ellipse's orientation aligns with the greenery's long axis, while in the buffer zone, it is governed by high-rise building shadows. This suggests that greenery within the waterless park primarily dictates the cold island intensity direction inside the park, whereas high-rise building distribution influences the cold island intensity direction within the buffer zone.

In summary, water bodies play a pivotal role in determining the direction of cold island intensities within watered parks and their buffer zones. However, the strength of this influence is also influenced by the water bodies' proportion. Conversely, in parks without water or with a low percentage of water bodies and their buffer zones, the direction of cold island intensity is influenced not only by greenery but also by the distribution of high-rise buildings. This finding aligns with previous studies [60,72].

## Equity analysis of the CIE and its optimization

There is evidence of inequity in the provision of cooling services by UPGS in Nanchang. A network analysis model was developed, using 71 UPGS with CIEs as cooling supply points and 3024 residential locations in Nanchang as demand points. The results revealed that 71.2% of residents could access cooling services in the UPGS's PCA within a 15-minute

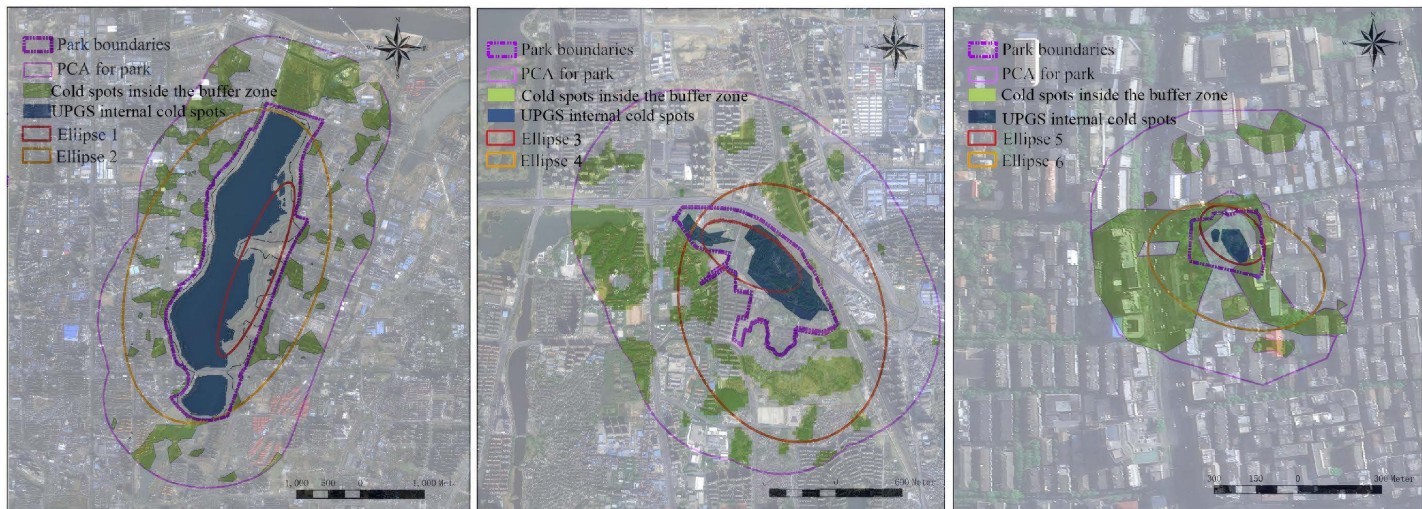

**Fig 9. Directional analysis of UPGS CIE: Aixi Lake Park (a), Meihu Park (b), Yangming Park (c) (The base map for this figure contains satellite imagery from the U.S. Geological Survey (public domain)).**

walk, while the remaining 28.8% had to endure at least a 15-minute walk in hot weather to access the same cooling services (Fig 10a, Table 2). Additionally, 10.9% of residents had the convenience of accessing this cooling service without leaving their homes, and 15.6%, 29.7%, and 25.9% of residents could access it within 5, 10, and 15 minutes, respectively.To further evaluate the spatial equity of cooling service accessibility, we conducted a quantitative analysis using the Coefficient of Variation (CV), Global Moran's I, and the Gini coefficient. The results indicate that the distribution of UPGS in central Nanchang exhibits a significant clustered pattern (CV = 167.96%, Moran's I = 0.087) and a high level of inequality in resource allocation among neighborhoods (Gini = 0.58). These quantitative metrics collectively substantiate the severe spatial inequity in the distribution of cooling services. Detailed visualizations and computational reports are provided in the Supplementary Material.

Fig 10a also illustrates the spatial disparity in the distribution of UPGS cooling services across various parts of the city. While 14% of residents within the First Ring Road were directly situated in the PCA of UPGS, only 8.6% and 3.8% of residents inside and outside the Second Ring Road had direct access to cooling services, respectively. Moreover, within a 5-minute and 10-minute walk, residents within the First Ring Road held a significant advantage over those in both the Second Ring Road and outside it. Within a 15-minute walk, 73.1% of residents in the First Ring Road and 72.5% of residents in the Second Ring Road could reach the PCA of UPGS, whereas only 59.1% of residents outside the Second Ring Road could access cooling services. In other words, nearly 190,000 suburban residents lacked access to UPGS-provided cooling services during extreme hot weather.

To maximize the chance that urban residents will be able to enjoy the cooling services provided by the UPGS within a 15-minute walk we have set the 612 settlements in the study area that require more than a 15-minute walk to reach the PCA as the target to be optimized. Subsequently, we utilized the KMS algorithm in Matlab to cluster these 612 settlements, determining the optimal number of UPGS by minimizing the Euclidean distance from each settlement to the cluster centroid to which it belonged. After several iterations, we obtained the K-means clustering curve for optimizing UPGS in Nanchang (Fig 11a). The horizontal axis of the curve represents the number of new UPGS, the vertical axis represents the average farthest distance from the settlement to be optimized to the clustering center, and the slope of the curve indicates the impact of increasing the number of clustering centers on clustering effectiveness. The clustering curve reveals that the slope gradually flattens out when the number of new UPGS reaches 18, signifying that further increasing the number of UPGS will not significantly enhance clustering effectiveness. Therefore, this study considers a total of 18 new UPGS, taking into account the cost and the number of people covered, to enhance cooling services in Nanchang.

The KMS algorithm addresses the issue of determining the number of new UPGSs but lacks specific location information for guiding actual planning. To overcome this limitation, we employed the general and robust PSO algorithm, recognized for its effectiveness in highly nonlinear and discontinuous scenarios [73].The PSO parameters were set following established standards in the literature [74]: a swarm size of 30 particles, an inertia weight of 0.729, and cognitive and social acceleration coefficients of 1.494. In the PSO algorithm implemented in Matlab, we utilized the spatial locations of the 18 UPGSs as the final output. Our optimization objective function combines the sum of the products of the minimum distances from settlements to the 18 particles and the maximum number of covered settlements. After nearly 2,000 model iterations, we obtained spatial information about the locations of the 18 new UPGSs (Fig 11b) (Fig 10b). To verify the robustness of the 18-site solution, the PSO algorithm was executed 20 times from different random initializations. This analysis confirmed that over 85% of the selected sites were consistently identified in more than 70% of the runs, demonstrating the stability and reliability of the optimization result. Of these, two new UPGSs were added within the first ring road, both situated at the boundary of the first ring road. Seven new UPGSs were introduced within the Second Ring Road, and the largest number of new UPGSs were placed outside the Second Ring Road, constituting half of the total new UPGSs. This further underscores the inequity of the UPGS CIE in Nanchang's spatial distribution.To quantitatively validate the PSO-derived optimization scheme, we simulated the post-optimization cooling service scenario. The 18 proposed UPGS were assigned a PCA of 2853 m, the optimal coverage radius identified by the K-means and PSO

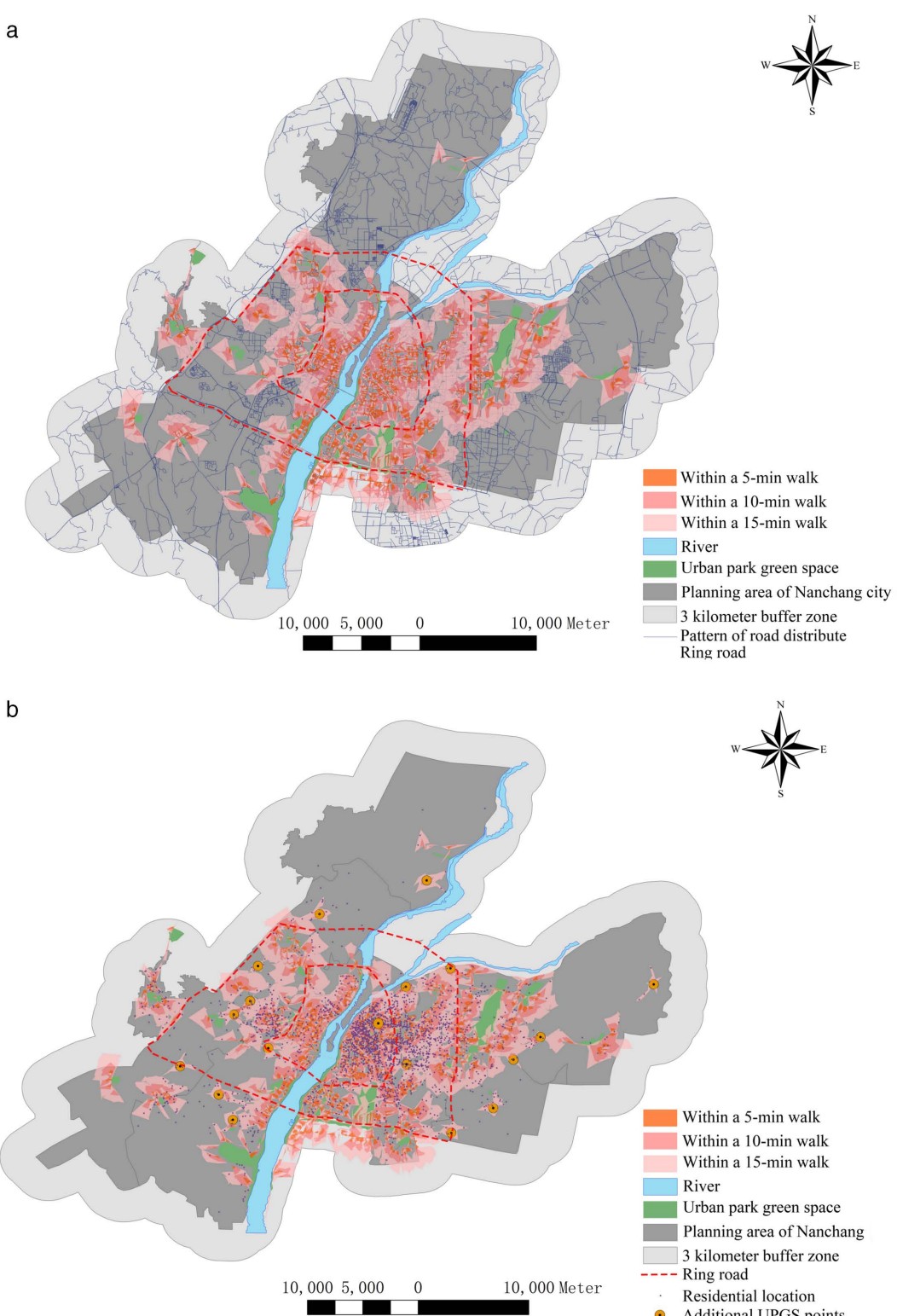

a

Within a 5-min walk
Within a 10-min walk
Within a 15-min walk
River
Urban park green space
Planning area of Nanchang city
3 kilometer buffer zone
Pattern of road distribute
Ring road

10,000 5,000 0 10,000 Meter

b

Within a 5-min walk
Within a 10-min walk
Within a 15-min walk
River
Urban park green space
Planning area of Nanchang
3 kilometer buffer zone
Ring road
Residential location
Additional UPGS points

10,000 5,000 0 10,000 Meter

**Fig 10.  (a) Spatial distribution of new UPGS points.** (b). Accessibility distribution of UPGS cooling services.

**Table 2. Accessibility results for residents in the UPGS cooling service area.**

|  | Total population | One ring region | Second ring region | Outside the second loop |
|---|---|---|---|---|
| **Accessibility level** | 3817614 | 2038154 | 1312476 | 466984 |
| **Within PCA** | 415900<br>10.9% | 284780<br>14.0% | 113182<br>8.6% | 17938<br>3.8% |
| **Within a 5-min walk** | 594696<br>15.6% | 374826<br>18.4% | 185096<br>14.1% | 34774<br>7.4% |
| **Within a 10-min walk** | 1132716<br>29.7% | 688422<br>33.8% | 327994<br>25.1% | 116300<br>24.9% |
| **Within a 15-min walk** | 990070<br>25.9% | 426124<br>20.9% | 438952<br>33.4% | 124994<br>26.8% |
| **> 15-min walk** | 1100132<br>28.8% | 548782<br>26.9% | 360434<br>27.5% | 190916<br>40.9% |

(Note: The top half of each row of the data in the table represents the number of people and the bottom half represents the percentage of the population.)

algorithms. Re-running the network analysis revealed a dramatic improvement in 15-minute walking accessibility: population coverage increased from 71.2% to 92.5%, an absolute gain of 21.3% (Fig 10b). This provides quantitative and visual confirmation that the strategy successfully extends cooling services to a larger population, effectively filling prior service gaps. It is important to note that in practical application, the area of new UPGS is flexible and service ranges are dynamic. Moreover, addressing cooling access in peripheral suburbs with poor road connectivity remains a challenge, underscoring the need for complementary strategies focused on enhancing the transportation network itself.

## 4. Discussion

### Factors affecting the cooling island effect of UPGS

Numerous studies have demonstrated that UPGS mitigates the UHI effect, and we also found that 71 out of 85 UPGS have a significant CIE in our study based on the CIE of UPGS in Nanchang City. The cooling characteristics were categorized into UPGS basic constituent indexes, surface cover indexes, and landscape pattern indexes for separate analysis.

Within the basic composition indicators of UPGS, it has been confirmed that the UPGS area exhibits a positive correlation with the CIE, and the cooling effect of UPGS strengthens with an increase in area [17]. However, the UPGS area threshold varies across different regions. Within this threshold, an increase in UPGS area significantly enhances the cooling island effect, but this enhancement effect gradually diminishes once the threshold is exceeded [75,76]. This study also confirms that the area threshold for the CIE of UPGS in Nanchang is 60 hm$^2$, which closely aligns with the results of a study on the area threshold for parks in Xiamen, China [76]. Additionally, compared to existing quantitative studies on the UPGS perimeter and its relationship with the CIE, the perimeter threshold for the optimal CIE determined in this study is 3 km. This is smaller than the 3.6 km threshold in Changzhou City, China, and larger than the 2.5 km threshold in Changchun City, China [77]. These variations likely arise from distinct regional climatic conditions and urban morphological contexts. Factors such as the average urban fabric density, prevailing wind patterns, and dominant vegetation types within UPGS can modulate how cooling effects scale with park size and shape, leading to these location-specific thresholds [78]. The relationship between the UPGS shape index and the CIE remains controversial. Some studies suggest that a larger surface area of contact between the UPGS and the external environment contributes more to the CIE [22]. In contrast, other studies argue that a larger contact surface increases the likelihood of heat exchange between the UPGS and the external environment. In this study, we found a negative correlation between the UPGS shape index and both LST and PCA. This indicates that a smaller contact surface between the UPGS and the external environment is more favorable

a

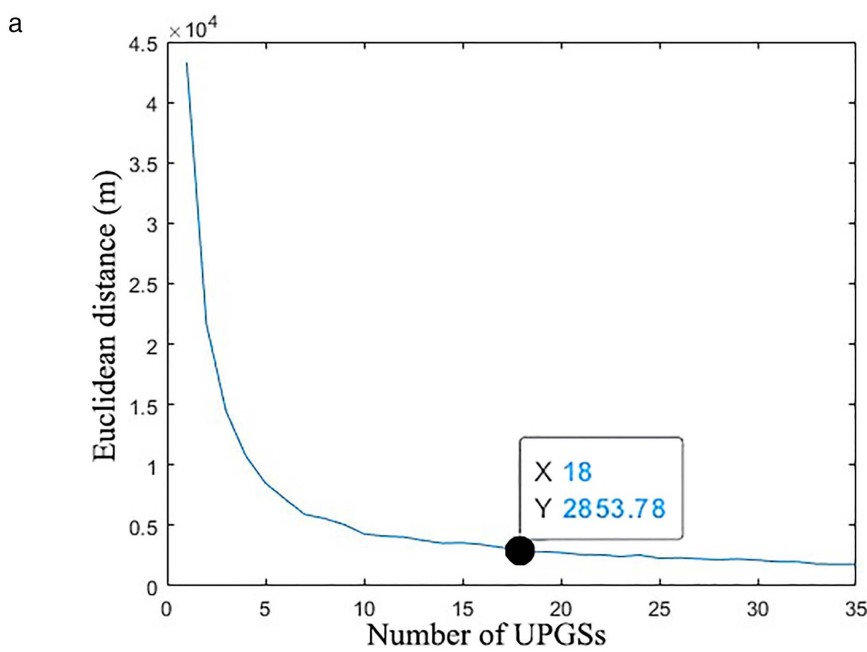

b

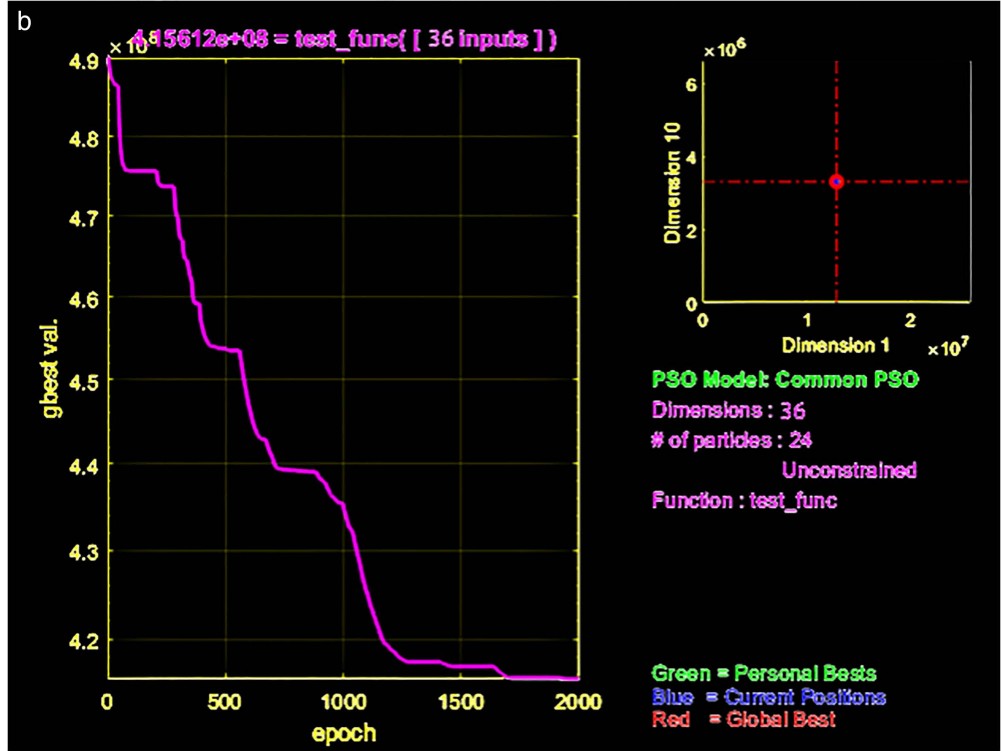

**Fig 11. (a) K-means clustering curve of UPGS.** (b) PSO algorithm iteration interface.

for the CIE. When combining the UPGS area, perimeter, and shape index, it becomes evident that a smaller shape index within the area and perimeter thresholds results in a stronger CIE.

The ground cover type within UPGS has varying effects on the CIE. In this study, impervious surface coverage was found to be negatively correlated with the CIE, meaning that lower impervious surface coverage results in smaller PCI and larger PCA in UPGS. This finding aligns with a study in a Mexican city [20] but contradicts a study in Changzhou City, China [77]. On the other hand, the positive correlation between water cover and the CIE of UPGS in this study has also been supported by the most relevant studies, which have concluded that water contributes the most to the CIE among the three cover categories [17,30]. Additionally, a crucial finding of this study is the absence of a statistically significant correlation between overall green coverage and the three quantitative CIE indicators. This suggests that in Nanchang's context, the mere presence of green area is not a reliable predictor of cooling efficiency. Several factors may explain this result. First, the "green coverage" in many UPGS of Nanchang may be dominated by less effective lawn areas [31], which possess a much lower cooling capacity compared to trees. Second, the resolution of Landsat thermal infrared data may be insufficient to capture the fine-scale cooling variations generated by different vegetation structures. Most importantly, our analysis of landscape pattern indices (Fig 8) revealed that the complexity and aggregation of green patches (LSI, AI) showed a stronger association with the cooling range (PCA) than green coverage itself. This implies that for enhancing the CIE, how green spaces are arranged and structured within a UPGS may be more determinant than how much green area is present.

Currently, quantitative research on landscape pattern indexes and their relationship with the CIE in UPGS has received limited attention. Determining the number of dimensions for landscape pattern indices can be challenging due to their diversity [58]. In this study, we introduced six categories of landscape pattern indices similar to those used in previous work and compared our findings with existing studies. We reached partially similar conclusions, revealing that lower impervious surface patch density, a smaller percentage of impervious surface coverage, and greater impervious surface aggregation are more favorable for the CIE in UPGS [48,63]. Moreover, a higher percentage of water body patches, increased separation, and finer-grained surface temperatures in UPGS were associated with lower temperatures. This could be explained by the greater number of water body patches with larger contact areas within UPGS, facilitating heat exchange and resulting in more significant cooling benefits [79]. Additionally, a higher degree of aggregation and more complex morphology of green space patches were found to enhance the range of temperature reduction in UPGS, consistent with a related study conducted in Beijing, China [70].

In this study, we also explored the direction of the CIE's intensity within parks by extracting highly representative UPGSs. The findings are valuable for guiding the directional layout of surface cover types within UPGSs and urban residential locations, optimizing the cooling services provided by UPGSs at the landscape composition indicator level. Our study revealed that water bodies within watered UPGSs play a crucial role in determining the directionality and intensity of the cooling island effect within parks and buffer zones. However, the strength of this effect is also influenced by the proportion of the UPGS area occupied by water bodies. In areas with a larger water body area, the directionality of the CIE strength aligns with the long axis of the water body, and this directionality is more pronounced. Conversely, in areas with small or no water bodies, the directionality can be influenced by green cover and high-rise building layouts. These findings are consistent with a previous study [60].

## Inequity in the distribution of the UPGS cooling island effect

There has been limited academic discussion on the equity of residents' access to the cooling range provided by UPGS, but this study highlights the unequal access to cooling services for urban residents. We assessed residents' accessibility to UPGS cooling services using network analysis and found that 71.2% of residents in the study area can reach the UPGS cooling range within a 15-minute walk, while 28.8% cannot access equivalent cooling services within the same timeframe. This inequality is particularly pronounced across different urban areas. Urban residents within the First Ring Road have

better access to cooling services than those in the Second Ring Road and much better access than those outside the Second Ring Road. For instance, 73.1% of urban residents within the First Ring Road and 71.3% within the Second Ring Road can access UPGS cooling services within a 15-minute walk, whereas nearly half of the residents outside the Second Ring Road lack access to the same level of cooling during extremely hot weather. Similar spatial accessibility inequities in cooling services have been observed in other cities worldwide [40,57,80].

The pronounced spatial inequity in cooling access is not a random outcome but is rooted in complex socio-urban processes. Similar to patterns observed in other cities [38,39], the historical trajectory of urban development in Nanchang has likely prioritized green infrastructure investment within the core urban areas, leading to a legacy of under-provision in newly developed or peripheral zones. Furthermore, the distribution of UPGS is closely intertwined with urban land economics, where higher land values in the city center can paradoxically both justify and constrain green space creation, while suburban expansion often occurs without commensurate investment in public amenities. Lastly, prevailing urban planning policies, which may have previously emphasized economic development over environmental justice, have played a critical role in shaping this unequal landscape.

## Optimization of the UPGS CIE and its distribution

This study focuses on optimizing the urban cooling island effect through a combined approach that considers both micro and macro perspectives. While existing research primarily emphasizes micro-level optimization of UPGS cooling effects by enhancing landscape pattern indicators [56], our approach is more comprehensive, encompassing a wider range. We summarize optimization strategies for the cooling island effect by analyzing the correlation between quantitative indexes of cooling and UPGS landscape composition indicators, including basic composition indexes, surface coverage, and landscape pattern indexes. Additionally, we explore the directionality of the UPGS CIE, which can help improve cooling services for UPGS residents.

Optimization from a macro perspective has been less frequently mentioned, and some studies have less frequently taken the residents' perspective and proposed specific practical guidance for the mitigation of urban thermal environments [80]. In this study, after revealing the unfairness of the CIE in urban UPGS through the network analysis method, and comparing PSO with other commonly used location-allocation optimization algorithms in related fields, namely the Genetic Algorithm (GA) and Ant Colony Optimization (ACO). The analysis revealed that while GA is a powerful global search tool, its performance is often highly dependent on the fine-tuning of operators (e.g., crossover and mutation rates) and it can be susceptible to premature convergence. ACO excels in combinatorial optimization problems; however, its computational load increases significantly when confronted with facility location problems characterized by a vast solution space [81]. In contrast, the PSO algorithm offers a simpler implementation process, requires fewer parameters to adjust, and exhibits faster convergence. This, coupled with its inherent synergy with K-means clustering, renders it more suitable for urban-scale planning scenarios.

## Research limitations and future directions

Beyond the research findings, this study has several limitations. Firstly, the limited resolution of the remote sensing image data obtained from Landsat 8TIRS introduces uncertainty when assessing the cooling effect of UPGS, particularly for smaller parks where the cooling contribution of key elements like water bodies or tree clusters might be obscured. In the future, supplementing remote sensing data with field temperatures measured by unmanned aerial vehicles equipped with infrared thermal cameras could improve accuracy. Secondly, this study relies on a single Landsat image from a late-summer day. While this captures a typical scenario of high heat stress, the snapshot nature of the data means our accessibility model represents a static picture. It does not account for seasonal variations in vegetation (e.g., reduced evapotranspiration in winter) or diurnal shifts in both park usage and cooling intensity. Future work should incorporate multi-temporal satellite imagery and ground measurements to investigate the dynamics of cooling services across different

timescales. Thirdly, while we have utilized population data specific to residential areas, we have not segmented the population based on socio-economic status or vulnerability. By treating the population as a homogeneous group, our equity analysis likely masks critical disparities. For instance, elderly or low-income residents, who are often more vulnerable to heat and have constrained mobility, might experience effective "access gaps" even if they reside within the nominal 15-minute service area [82]. Our Gini coefficient and coverage rate, therefore, represent a baseline of spatial equity that may not fully reflect the experienced inequity among different social groups. Additionally, our focus has been solely on assessing the equity of UPGS cooling access for urban residents, without considering variations in cooling effectiveness among different UPGSs, potentially leading to supply-demand imbalances [83]. Lastly, there remains considerable arbitrariness in determining the locations of additional UPGSs, and we have not precisely accounted for supply pressure and construction area considerations for each additional UPGS.

## 5. Conclusion

This study investigated the factors influencing the UPGS CIE and developed strategies for the critical but often overlooked issue of equitable access to this service in Nanchang, China.We quantified and analysed the correlation between the cooling island effect of UPGS and landscape composition indicators in Nanchang, China, and proposed a strategy to optimise the cooling island effect of UPGS under a micro perspective. We found that UPGS with an area of 60 hm$^2$ and a perimeter of 3 km can produce an efficient CIE; increasing water body coverage and decreasing the impervious surface coverage can reduce surface temperature and expand the cooling range; reducing the density and percentage of impervious surface patches, increasing the percentage and separation of water body patches, and increasing the degree of aggregation and morphological complexity of green patches can effectively reduce the UPGS's CIE. The direction of the UPGS cold island intensity is strongly influenced by the area of the water body, the direction of the long axis of the water body, and the distribution of high-rise buildings. In addition, we quantified the accessibility of urban residents to the cooling range of the UPGS by network analysis and found that the level of cooling services enjoyed by urban residents within the First Ring Road is greater than that of residents in the Second Ring Road and much greater than that of urban residents outside the Second Ring Road. Nearly half of the urban residents outside the second ring road do not enjoy cooling services from the UPGS within a 15-minute walk. To address this inequity, we used a combination of the KMS algorithm and the PSO algorithm to calculate the spatial point information of 18 UPGSs that take into account the cost and the maximum coverage. This strategy increases the population coverage within a 15-minute walk from 71.2% to 92.5% and optimizes the spatial pattern of the urban UPGS cooling island effect from a macro perspective. Theoretically bridging a critical gap in cooling equity research, this work practically enables enhanced thermal resilience through spatially optimized UPGS planning.

## Supporting information

**S1 File. Supplementary material.**
(ZIP)

**S2 File. Renamed_1fc36.**
(ZIP)

## Author contributions

**Conceptualization:** YouQiang Zhao.

**Data curation:** Liu Ping yi.

**Funding acquisition:** Peng Gong.

**Investigation:** YouQiang Zhao.

**Resources:** Zhang jian ping.

**Software:** YouQiang Zhao.

**Supervision:** Peng Gong.

**Validation:** YouQiang Zhao, Liu Ping yi.

**Visualization:** Liu Ping yi.

**Writing – original draft:** YouQiang Zhao.

**Writing – review & editing:** Peng Gong, Zhang jian ping.

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
