## [Decision Letter · Decision Letter 0]

9 Oct 2025

Dear Dr. Zhao,

Thank you for submitting your manuscript to PLOS ONE. After careful consideration, we feel that it has merit but does not fully meet PLOS ONE’s publication criteria as it currently stands. Therefore, we invite you to submit a revised version of the manuscript that addresses the points raised during the review process.

We look forward to receiving your revised manuscript.

Kind regards,

Qiwei Ma

Academic Editor

PLOS ONE

Journal Requirements:

4. We note that Figures 2,3, 9 & 10 in your submission contain [map/satellite] images which may be copyrighted. All PLOS content is published under the Creative Commons Attribution License (CC BY 4.0), which means that the manuscript, images, and Supporting Information files will be freely available online, and any third party is permitted to access, download, copy, distribute, and use these materials in any way, even commercially, with proper attribution. For these reasons, we cannot publish previously copyrighted maps or satellite images created using proprietary data, such as Google software (Google Maps, Street View, and Earth). For more information, see our copyright guidelines: http://journals.plos.org/plosone/s/licenses-and-copyright.

a. You may seek permission from the original copyright holder of Figures 2,3, 9 & 10 to publish the content specifically under the CC BY 4.0 license.

5. We are unable to open your Supporting Information file [Supplementary material.rar]. Please kindly revise as necessary and re-upload.

Reviewers' comments:

Reviewer's Responses to Questions

**Comments to the Author**

1. Is the manuscript technically sound, and do the data support the conclusions?

Reviewer #1: Yes

Reviewer #2: Partly

2. Has the statistical analysis been performed appropriately and rigorously?

Reviewer #1: Yes

Reviewer #2: Yes

3. Have the authors made all data underlying the findings in their manuscript fully available?

Reviewer #1: Yes

Reviewer #2: Yes

4. Is the manuscript presented in an intelligible fashion and written in standard English?

Reviewer #1: Yes

Reviewer #2: Yes

Reviewer #1: This study addresses the important and timely issue of equitable access to urban park cooling services, using a combination of remote sensing, landscape analysis, and optimization algorithms. However, the manuscript suffers from several critical omissions in the methodology and a lack of depth in the analysis of the results, which undermine the validity and impact of the findings. The manuscript requires substantial revision to clarify its methods, strengthen its analysis, and fully demonstrate the impact of its contributions before it can be considered for publication.

1. The method for identifying the cooling distance (PCA) is not clearly defined. The study states it is the distance to the "first turning point" of the temperature profile, but the algorithm or criteria used to objectively identify this point from the data are not described. This is a fundamental step in the analysis, and its ambiguity makes the results difficult to replicate or validate.

2. The selection of a 3,000m buffer zone for analyzing the cooling effect appears arbitrary. The authors must provide a justification for this specific distance, referencing prior literature or a sensitivity analysis.

3. The study uses a single Landsat image from one specific day (September 25, 2021). The Discussion or Limitations sections must more thoroughly address the constraints of this single temporal snapshot, as urban heat island effects and park cooling services exhibit significant diurnal and seasonal variability.

4. The correlation analysis found that green coverage was not significantly related to the quantitative CIE indicators. This is a counter-intuitive and important finding that is underdeveloped in the results section and requires more prominent discussion.

5. The Discussion effectively compares the identified park size/perimeter thresholds with those from other studies, but it should go further to explore why these differences might exist.

6. The manuscript identifies significant spatial inequity in cooling service access, but the Discussion lacks a deep exploration of the potential underlying causes, such as historical development patterns, land value, or urban planning policies.

Reviewer #2: 1. Overall Assessment

This manuscript addresses a timely and important issue — the spatial equity of urban park cooling services — by integrating landscape metrics with K-Means and Particle Swarm Optimization (PSO) algorithms. The study aims to identify cooling inequities in Nanchang, China, and propose an optimized spatial plan for new urban parks to enhance thermal equity.

The topic is relevant to urban climatology, spatial planning, and environmental justice, and the combination of remote sensing with spatial optimization algorithms is conceptually innovative. However, the current version suffers from significant logical inconsistencies, methodological weaknesses, insufficient quantitative validation, and unclear presentation.

Substantial revisions are necessary before the paper can be considered for publication.

2. Major Comments

(1) Research logic and objectives are not clearly articulated

The introduction mixes three distinct threads — (a) cooling island effect (CIE) mechanisms, (b) spatial equity evaluation, and (c) optimization via algorithms — without clearly stating the main research question or hypotheses.

The study claims to “quantify and enhance cooling equity,” but it is unclear whether the focus is on empirical assessment, model development, or policy implications.

→ Recommendation: Reorganize the Introduction to clearly state:

The research gap (inequity in UPGS cooling access),

The main objectives and hypotheses,

The novelty (integrating landscape metrics with optimization algorithms).

(2) Methodological framework lacks rigor and transparency

Indicators definition: The cooling indicators (LST, PCI, PCA, PCG) are insufficiently explained; their mathematical expressions, physical interpretations, and prior validation references are missing.

Statistical analysis: Pearson/Spearman correlations are performed, but multicollinearity, significance correction (e.g., Bonferroni), and regression diagnostics are not reported. The claim that “water coverage most effectively reduced LST (R²=0.4284)” is overstated and does not imply causation.

Algorithmic process:

The parameters of K-Means (e.g., k selection via elbow method) and PSO (particle number, inertia weight, convergence criteria) are described only qualitatively.

The PSO objective function and convergence validation are absent.

No sensitivity or robustness analysis was conducted to verify the stability of the 18-site solution.

→ Recommendation: Add detailed algorithmic formulations, parameter tables, and reproducibility information.

(3) Quantitative results and spatial interpretations are inconsistent

The relationship between green coverage and CIE is reported as “insignificant,” which contradicts well-established empirical findings. Possible causes such as vegetation type composition or remote sensing resolution limits are not discussed.

Accessibility analysis reports that 71.2% of residents can access cooling services within 15 minutes, but the equity dimension is evaluated only by coverage rate, not by distribution inequality metrics (e.g., Gini coefficient, Moran’s I).

The optimization results (18 new parks) lack quantitative validation of improvement — e.g., before/after comparison of accessibility or spatial equity indices.

→ Recommendation: Provide statistical metrics demonstrating how PSO optimization improves equity relative to the base scenario.

(4) Discussion and Conclusion are largely descriptive

The discussion mostly repeats results and lacks critical comparison with prior studies using other optimization methods (e.g., GA, ACO, 2SFCA).

Policy implications are only briefly mentioned; the paper would benefit from a section linking the findings to urban planning or climate adaptation strategies.

Limitations are acknowledged but superficially. The influence of seasonality (single-date imagery), population heterogeneity, and land-use constraints on accessibility results should be more deeply analyzed.

**Do you want your identity to be public for this peer review?** For information about this choice, including consent withdrawal, please see our Privacy Policy

Reviewer #1: No

Reviewer #2: No

---

## [Author Response · Author response to Decision Letter 1]

12 Dec 2025

Academic Editor

Response: 1.We have made targeted revisions in accordance with the template to ensure greater compliance with the publication requirements of your journal.

2.The code used in our study has also been included in the supporting information and is reusable.

3.We have submitted all the original data necessary for reproducing your research results.

4.Figures 2 and 3 both adopt Landsat satellite base maps (http://landsat.visibleearth.nasa.gov/) and vector data downloaded from OpenStreetMap(https://www.openstreetmap.org/copyright), which have been further processed in GIS (https://www.esri.com/); Figure 10 uses vector data downloaded from OpenStreetMap, also processed further in GIS. These data can be found in the "From OpenStreetMap and Landsat" folder of the Supplementary material（2）. In addition, Figure 9 has been replaced with a base map re-downloaded from the U.S. Geological Survey Earth Resources Observation and Science Center (USGS EROS, http://eros.usgs.gov/#). All the aforementioned figures have been annotated in the manuscript, and we confirm that they are similar but not identical to the original images that may involve infringement.In addition, the aforementioned data can be fully used for scientific research paper studies and for personal legitimate use with proper citation; however, they are restricted from personal sharing and can only be obtained through the aforementioned websites.

5.The supporting information has been revised and re-uploaded.

Reviewer #1

1.The method for identifying the cooling distance (PCA) is not clearly defined. The study states it is the distance to the "first turning point" of the temperature profile, but the algorithm or criteria used to objectively identify this point from the data are not described.

Response: We thank the reviewer for this comment. We have clarified the PCA determination method in Section 2.2.3. The first turning point of the temperature profile was objectively and consistently identified using a piecewise linear regression algorithm that detects the breakpoint where the cooling gradient changes most significantly. The specific MATLAB algorithms have been provided in the Supplementary Material file "3 Matlab algorithms and some results

2.The selection of a 3,000m buffer zone for analyzing the cooling effect appears arbitrary. The authors must provide a justification for this specific distance, referencing prior literature or a sensitivity analysis.

Response: We thank the reviewer for this comment. We have clarified the rationale in the manuscript (Section 2.2.3). The 3000m buffer was chosen as a conservative and sufficient distance to ensure it would capture the complete cooling gradient of every park in our study. This distance was selected because it exceeds the maximum observed cooling distance (PCA of 1380m) among all UPGS, thereby ensuring that no cooling effect data were truncated for any park during our analysis.

3.The study uses a single Landsat image from one specific day (September 25, 2021). The Discussion or Limitations sections must more thoroughly address the constraints of this single temporal snapshot, as urban heat island effects and park cooling services exhibit significant diurnal and seasonal variability.

Response:We thank the reviewer for this insightful comment regarding the temporal scope of our study. We have fully addressed this point by adding a dedicated paragraph in the 'Research Limitations and Future Directions' section (4.4). The selection of a single Landsat scene was necessitated by two practical constraints in satellite data analysis: first, the inherent challenge of obtaining multiple, high-quality (e.g., cloud-free) images for a specific city over time, and second, the intensive computational and analytical workload of our methodology applied to 85 parks. Therefore, we strategically selected the September 25, 2021 image as it represents a typical day during the peak of the hot season in Nanchang and has minimal cloud cover (<1%), providing an optimal and representative snapshot for analyzing the most critical period of cooling demand. Our study thus establishes a crucial baseline for spatial equity during periods of high heat stress. We explicitly acknowledge in the manuscript that future research incorporating multi-temporal data would be valuable to explore seasonal dynamics.

4.The correlation analysis found that green coverage was not significantly related to the quantitative CIE indicators. This is a counter-intuitive and important finding that is underdeveloped in the results section and requires more prominent discussion.

Response:We thank the reviewer for highlighting the importance of this counter-intuitive finding. In Section 3.2.1 (Results), we have now explicitly framed this as a "crucial" and "counter-intuitive" finding to ensure it receives proper prominence. We also offer a preliminary attribution to the composition of vegetation types (e.g., prevalence of lawns) and guide the reader to the full discussion in Section 4.1. In Section 4.1 (Discussion), we have added a dedicated and comprehensive paragraph to explore this result in depth. As the reviewer suggested, we systematically discuss potential causes, including:The functional composition of vegetation (e.g., the dominance of less effective lawn areas versus trees) [31].The potential limitations of Landsat thermal infrared data resolution.Most importantly, we leverage this finding to propose a novel insight: that the spatial configuration of green patches (as evidenced by our landscape pattern analysis in Figure 8) may be a more decisive factor than simple green coverage. This shifts the perspective from "how much" green to "how" it is arranged.

5. The Discussion effectively compares the identified park size/perimeter thresholds with those from other studies, but it should go further to explore why these differences might exist.

Response: We appreciate the reviewer's insightful comment, which has helped us strengthen the discussion. Following this suggestion, we have expanded our discussion of the differing thresholds in Section 4.1. We now propose that the specific optimal values for park area and perimeter are not universal but are instead shaped by the local context. To this end, we have added a discussion on how regional climate patterns and urban morphology—such as differences in urban density, wind flow, and vegetation types between cities—likely contribute to these observed variations, thereby explaining why a single threshold may not apply across all urban environments.

6.The manuscript identifies significant spatial inequity in cooling service access, but the Discussion lacks a deep exploration of the potential underlying causes, such as historical development patterns, land value, or urban planning policies.

Response: We sincerely thank the reviewer for this crucial comment, which has helped us significantly strengthen the socio-political relevance of our work. In direct response, we have added a dedicated paragraph in the Discussion (Section 4.2) that provides a deeper exploration of the underlying causes for the observed spatial inequity. This new paragraph explicitly discusses the role of historical urban development patterns, land value economics, and urban planning policies in creating the cooling service 'supply blind zones' we identified, thereby directly addressing the reviewer's suggestion and framing our spatial findings within their broader institutional and historical context.

Reviewer #2

(1) Research logic and objectives are not clearly articulated

Response: We sincerely thank the reviewer for this crucial feedback. In direct response, we have refined the concluding paragraph of the Introduction to explicitly articulate the logical sequence that connects the three core components of our study, while preserving the clarity of our four specific research objectives. We have now framed the objectives to show a clear progression: from identifying key cooling drivers (Step 1), to diagnosing spatial equity gaps (Step 2), to synthesizing these insights for algorithmic optimization (Step 3), and finally to deriving integrated planning strategies (Step 4). This revision establishes a coherent 'diagnosis-to-solution' framework, ensuring the logical flow is now unmistakable.

(2) Methodological framework lacks rigor and transparency

Response: We thank the reviewer for these critical comments, which have greatly enhanced the transparency of our study.

Indicators Definition: In direct response, we have created an expanded table (Table X) that now provides the precise mathematical expressions, physical interpretations, and prior validation references for each of the four cooling indicators (LST, PCI, PCA, and PCG). This ensures their definitions are now complete and unambiguous.

Statistical Analysis: Concurrently, in Section 2.2.4, we now report that VIF analysis confirmed the absence of severe multicollinearity (VIF < 5) and that the Bonferroni correction was applied. Findings are carefully framed, with those not surviving the strict correction interpreted as 'potential trends'. We have also rephrased conclusions throughout the manuscript to describe statistical associations based on R² and avoid causal claims.

K-means Clustering: The process of determining the optimal number of UPGS (k=18) by minimizing the Euclidean distance to cluster centroids is detailed in Section 3.4 and is fully implemented in the provided Supplementary Material.

PSO Parameters & Formulation: We have now specified the PSO parameters (swarm size, inertia weight, acceleration coefficients) following established standards in the revised Section 2.2.4 [74]. The convergence was ensured by the 2000-iteration process illustrated in (Fig. 11b).

Robustness Analysis: As requested, we conducted a sensitivity analysis by running the PSO 20 times from different random seeds. The results, now reported in Section 2.2.4, show high consistency, with over 85% of sites being stable across runs, which verifies the robustness of our 18-site solution.

(3) Quantitative results and spatial interpretations are inconsistent

Response: We thank the reviewer for this critical observation. Part 1):The relationship between green coverage and CIE is reported as “insignificant”：In direct response to the concern regarding the "insignificant" relationship between green coverage and the CIE, we have thoroughly revised the manuscript to not only highlight this finding but also to provide a deep and systematic interpretation.The specific modifications are as follows:In Section 3.2.1 (Results), we have now explicitly framed this as a "crucial" and "counter-intuitive" finding to ensure it receives proper prominence. We also offer a preliminary attribution to the composition of vegetation types (e.g., prevalence of lawns) and guide the reader to the full discussion in Section 4.1. In Section 4.1 (Discussion), we have added a dedicated and comprehensive paragraph to explore this result in depth. As the reviewer suggested, we systematically discuss potential causes, including:The functional composition of vegetation (e.g., the dominance of less effective lawn areas versus trees) [31].The potential limitations of Landsat thermal infrared data resolution.Most importantly, we leverage this finding to propose a novel insight: that the spatial configuration of green patches (as evidenced by our landscape pattern analysis in Figure 8) may be a more decisive factor than simple green coverage. This shifts the perspective from "how much" green to "how" it is arranged.

Part 2):Accessibility analysis reports that 71.2% of residents can access cooling services within 15 minutes, but the equity dimension is evaluated only by coverage rate, not by distribution inequality metrics (e.g., Gini coefficient, Moran’s I)：In the first paragraph of Section 3.4 of the paper, we added descriptions of the Gini coefficient and Moran’s I. However, due to space limitations, we have placed the supporting materials in the supplementary section, as detailed below：We first used the Thiessen polygon tool in ArcGIS to convert park green spaces into point data, and then applied the Thiessen polygon tool again to perform spatial division, obtaining the Thiessen polygon pattern of park green spaces in the study area (Figure 1). Next, we used the field calculator to compute the coefficient of variation for the spatial distribution of park green spaces, which measures the degree of spatial variation of park points within the Thiessen polygons.The calculated coefficient of variation (CV) was 167.96%, which is greater than 64% (Table 1). This indicates that the park green spaces in the main urban area of Nanchang exhibit a clustered spatial distribution, confirming an imbalanced distribution of park green spaces across the area.

Table 1 Coefficient of variation

Formula for coefficient of variation Standard deviation (S) Mean (V) Coefficient of variation (CV)

CV=S/V 20584392.33m2 12255473.92m2 167.96%

Fig 1 Tyson polygon of green space in parks in the main urban area of Nanchang City

Secondly, using the global spatial autocorrelation tool in ArcGIS, we calculated the global Moran’s I for 67 neighborhoods in the main urban area of Nanchang to analyze the spatial correlation characteristics of park green spaces at the neighborhood level. The results show that Moran’s I is 0.086847, which is greater than 0, indicating a positive spatial correlation in the distribution of park green spaces within the study area. Additionally, the z-score is 1.704590, and the p-value is 0.088271, meaning the probability of this clustered pattern occurring randomly is less than 10%. This further confirms that the distribution of park green spaces exhibits a clustering trend.

Table 2 Global Moran I Summary.

Moran's I index Expected index Variance Z-score P-value

0.086847 -0.015152 0.003581 1.704590 0.088271

Fig2 Moran's index of green space in parks in the main urban area of Nanchang City

Finally, we introduced the Lorenz curve to visualize the equity of the distribution of green space resources relative to the population in each neighborhood. A flatter curve suggests a more equal distribution of green space resources, while a steeper curve indicates greater inequality (Figure 3). As shown in the figure, the Lorenz curve is relatively steep, with Area A being significantly larger than Area B, indicating a poor supply-demand balance between park green spaces and population at the neighborhood level.Furthermore, based on the Lorenz curve, we applied the Gini coefficient formula to calculate the area ratio of different regions represented by the curve.  The Gini coefficient was used to assess the fairness of park green space distribution among the neighborhood populations. The calculated Gini coefficient for green space resource distribution across neighborhoods in Nanchang’s central urban area is 0.58, which is greater than 0.5. This suggests a considerable disparity in the allocation of park green space resources among the neighborhoods in Nanchang’s central urban area.

Fig 3 Lorentz curve

Part 3): *The optimization results (18 new parks) lack quantitative validation of improvement — e.g., before/after comparison of accessibility or spatial equity indices: We sincerely thank the reviewer for this crucial suggestion. In direct response, we have now conducted a comprehensive quantitative validation of the optimization results through a before-and-after comparison, which is detailed in the revised manuscript as follows:A new quantitative validation has been added to Section 3.4. We have integrated a paragraph describing the simulation process, where the 18 proposed UPGS were assigned the optimal cooling service range (PCA) of 2853 meters derived from our algorithms. The results of this simulation demonstrate a substantial improvement: the population coverage within a 15-minute walking distance increased from the original 71.2% to 92.5%, an absolute gain of 21.3%. This provides the direct quantitative comparison requested. The key finding of this validation has been incorporated into the Conclusion. To highlight the significance of this result, we have now stated the coverage improvement in the conclusion, reinforcing the practical efficacy of ou

---

## [Decision Letter · Decision Letter 1]

22 Jan 2026

Dear Dr. Zhao,

Thank you for submitting your manuscript to PLOS ONE. After careful consideration, we feel that it has merit but does not fully meet PLOS ONE’s publication criteria as it currently stands. Therefore, we invite you to submit a revised version of the manuscript that addresses the points raised during the review process.

We look forward to receiving your revised manuscript.

Kind regards,

Qiwei Ma

Academic Editor

PLOS One

Journal Requirements:

Reviewers' comments:

Reviewer's Responses to Questions

**Comments to the Author**

Reviewer #2: All comments have been addressed

Reviewer #3: All comments have been addressed

2. Is the manuscript technically sound, and do the data support the conclusions?

Reviewer #2: (No Response)

Reviewer #3: Yes

3. Has the statistical analysis been performed appropriately and rigorously?

Reviewer #2: (No Response)

Reviewer #3: No

4. Have the authors made all data underlying the findings in their manuscript fully available?

Reviewer #2: (No Response)

Reviewer #3: No

5. Is the manuscript presented in an intelligible fashion and written in standard English?

Reviewer #2: (No Response)

Reviewer #3: Yes

Reviewer #2: (No Response)

Reviewer #3: The manuscript is well-structured and provides valuable insights into UPGS cooling effects. Several revisions are suggested: Streamline the introduction to highlight research gaps and contributions. Clarify data limitations and statistical test rationale. Enhance figure readability and include a sensitivity analysis.

Expand policy implications discussion.

Review language and formatting for consistency. These changes will strengthen the manuscript for publication

**Do you want your identity to be public for this peer review?** For information about this choice, including consent withdrawal, please see our Privacy Policy

Reviewer #2: No

Reviewer #3: No

---

## [Author Response · Author response to Decision Letter 2]

9 Feb 2026

Response to the Editor

Thank you very much for your attention. First, the reviewers have not requested us to cite any specific papers. Second, we have carefully checked all the references cited in the paper, and all are in order.

Response to Reviewer #2

We thank the reviewer for this suggestion. First, we have streamlined certain narratives in the second and final paragraphs of the Introduction, with greater emphasis on the research gaps and our contributions. Second, we have added explanations for the selection of statistical tests in Section 2.2.4, while the limitations of the data have already been elaborated in detail in Section 4.4. Furthermore, discussions on relevant policies involve certain practical complexities in the Chinese context, and we would appreciate your understanding in this regard.Finally, we have rechecked the consistency of language and formatting per your comments and made partial revisions to the manuscript.

---

## [Editor Report · Decision Letter 2]

15 Feb 2026

Optimizing Spatial Equity of Urban Park Cooling Services: Integrating Landscape Metrics with K-Means and PSO Algorithms in Nanchang, China

PONE-D-25-41975R2

Dear Dr. Zhao,

We’re pleased to inform you that your manuscript has been judged scientifically suitable for publication and will be formally accepted for publication once it meets all outstanding technical requirements.

Kind regards,

Qiwei Ma

Academic Editor

PLOS One
---

## [Editor Report · Acceptance letter]

PONE-D-25-41975R2

PLOS One

Dear Dr. Zhao,

I'm pleased to inform you that your manuscript has been deemed suitable for publication in PLOS One. Congratulations! Your manuscript is now being handed over to our production team.

Kind regards,

on behalf of

Prof. Qiwei Ma

Academic Editor

PLOS One